# Quantifying Uncertainty in Error Consistency: Towards Reliable Behavioral Comparison of Classifiers

**Thomas Klein**[1,2,3,4*]  **Sascha Meyen**[2]

**Wieland Brendel**[‡1,2,3,4]  **Felix A. Wichmann**[‡2]  **Kristof Meding**[‡2,5]

## Abstract

Benchmarking models is a key factor for the rapid progress in machine learning (ML) research. Thus, further progress depends on improving benchmarking metrics. A standard metric to measure the behavioral alignment between ML models and human observers is *error consistency* (EC). EC allows for more fine-grained comparisons of behavior than other metrics such as accuracy, and has been used in the influential Brain-Score benchmark to rank different DNNs by their behavioral consistency with humans. Previously, EC values have been reported without confidence intervals. However, empirically measured EC values are typically noisy—thus, without confidence intervals, valid benchmarking conclusions are problematic. Here we improve on standard EC in two ways: First, we show how to obtain confidence intervals for EC using a bootstrapping technique, allowing us to derive significance tests for EC. Second, we propose a new computational model relating the EC between two classifiers to the implicit probability that one of them copies responses from the other. This view of EC allows us to give practical guidance to scientists regarding the number of trials required for sufficiently powerful, conclusive experiments. Finally, we use our methodology to revisit popular NeuroAI-results. We find that while the general trend of behavioral differences between humans and machines holds up to scrutiny, many reported differences between deep vision models are statistically insignificant. Our methodology enables researchers to design adequately powered experiments that can reliably detect behavioral differences between models, providing a foundation for more rigorous benchmarking of behavioral alignment.

## 1 Introduction

Consider the following problem: *Given two classifiers operating on the same domain, how should we quantify their similarity?* This abstract problem has become highly practically relevant in the context of cognitive science and beyond. In cognitive science, deep neural networks (DNNs) have been proposed as models of the human visual system [Doerig et al., 2023, Kriegeskorte, 2015, Cichy and Kaiser, 2019, Kietzmann et al., 2017, Yamins et al., 2014], and thus the question arises how similarly they behave to human observers. In other domains, e.g. law or medicine, it is also crucial to know whether DNNs interpret and judge information similarly to humans. Thus, a metric is needed that can reliably quantify the degree of behavioral similarity between DNNs and human decision makers.

The standard method proposed for this purpose is *error consistency* (EC) [Geirhos et al., 2020], which has seen wide application in the context of human-machine comparisons, e.g. in Geirhos et al. [2021],

---

[*1] Max Planck Institute for Intelligent Systems, Tübingen, Germany [2] University of Tübingen. [3] Tübingen AI Center [4] ELLIS Institute Tübingen. [5] CSZ Institute for Artificial Intelligence and Law. [‡] Joint supervision. Correspondence to: t.klein@uni-tuebingen.de. Code: github.com/wichmann-lab/error_consistency.

39th Conference on Neural Information Processing Systems (NeurIPS 2025).

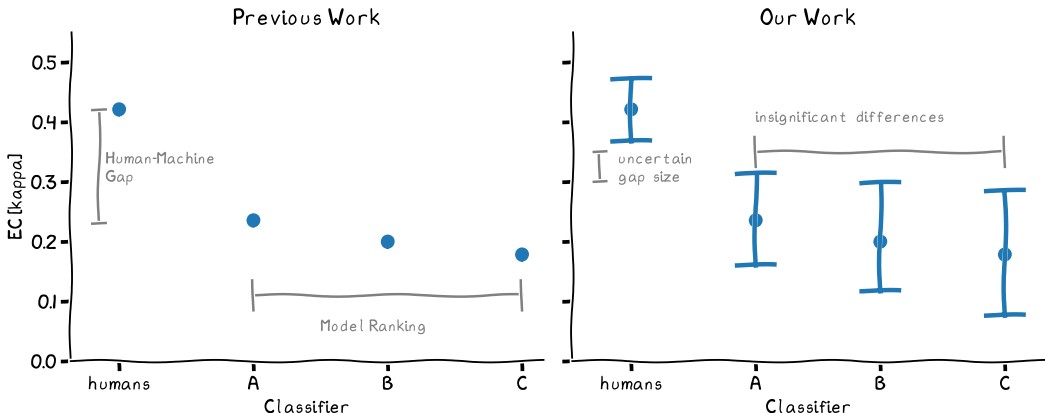

Figure 1: **Schematic: Error Consistency is noisy.** The gist of our findings is that by calculating confidence intervals for empirical EC values, we show that while the human-machine gap is probably real, the measure is not stable enough to resolve differences between models.

Ollikka et al. [2024], Li et al. [2025], Parthasarathy et al. [2023]. Similarity metrics based on EC have also seen application in the behavioral similarity section of the vision-branch of Brain-Score [Schrimpf et al., 2018]. More recently, a variation of EC—sharing many of the relevant properties, so that our considerations apply—has been proposed as a metric to compare Large Language Models (LLMs) [Goel et al., 2025].

EC builds on Cohen's $\kappa$ [Cohen, 1960], which considers the trial-level agreement observed for a pair of classifiers to compute a scalar similarity score, which is $1$ if the two models are behaviorally indistinguishable and $-1$ if they are as different as they could possibly be. A value of $0$ indicates independence. In practice, EC is reported as a point estimate of a unit-less scalar value, based on classification decisions obtained on a single set of test stimuli, like in Figure 2a. However, since EC relies on trial-by-trial similarity, the measure is inherently noisy, as individual trials can have a large impact on the final score, especially when the total number of trials is small—as is often the case when conducting experiments with humans.

Our contributions are as follows:

- We explain how confidence intervals for empirical EC values can be estimated on any dataset on which EC may be calculated using bootstrapping [Efron and Tibshirani, 1994].

- We propose a novel method of modeling classifier consistency, allowing us to simulate classifiers with a specific consistency and thus to plan sufficiently powerful, conclusive experiments.

- We apply our methods to two influential benchmarks (Model-vs-Human [Geirhos et al., 2021] and Brain-Score [Schrimpf et al., 2018]), showing that the currently available amount of human reference data is not sufficient to draw conclusions about model differences in many cases, as illustrated in Figure 1.

- We provide a python package containing utility functions for obtaining confidence intervals around error consistency values, calculating p-values, planning experiments, and testing results for significance.

We will begin by providing a brief introduction to EC in Section 2, to then explain how confidence intervals can be calculated in Section 3. A detailed Related Work section is provided in Appendix A since we discuss related work on measuring behavioral alignment, Error Consistency, and benchmarking.

## 2 Error Consistency

First, we provide a brief introduction to error consistency and introduce our notation; see Geirhos et al. [2020] for an in-depth description. EC is measured by applying Cohen's $\kappa$ to evaluate whether two classifiers' responses are jointly correct or incorrect—that is, whether they are consistent about when they make errors. The experimental setting in which EC can be calculated is the following: Two classifiers categorize each of $N$ samples (e.g., natural images) into one of $K$ classes (e.g., the 1,000 ImageNet classes). Each sample, $x_i$, has a ground-truth label $y_i \in \{1, ..., K\}$ and receives classifications, $\hat{y}_i^{(j)} \in \{1, ..., K\}$, with $j \in \{1, 2\}$. Whether these classifications are correct or not is coded as $r_i^{(j)} \in \{0, 1\}$ where $r_i^{(j)} = 1$ if and only if the $j$-th classifier responded with the true label, $\hat{y}_i^{(j)} = y_i$. This renders EC agnostic to the number of classes of the classification task because it only considers agreement on which samples are jointly classified correctly and incorrectly.

In this setting, EC is measuring the agreement between $r^{(1)}$ and $r^{(2)}$ using Cohen's $\kappa$ (which is typically applied to $\hat{y}^{(1)}$ and $\hat{y}^{(2)}$):

$$\kappa = \frac{p_{obs} - p_{exp}}{1 - p_{exp}} \tag{1}$$

where $p_{obs}$ is the observed agreement, that is, the proportion of trials which are jointly classified correctly or incorrectly. This quantity is normalized by the amount of agreement expected from independent classifiers, $p_{exp}$. This normalization accounts for the fact that classifiers with high accuracies are a priori expected to agree more often in their classifications than classifiers with low accuracies.

However, the normalization does not render EC fully orthogonal to the accuracies of the classifiers, because Cohen's $\kappa$ depends on the marginal distributions of classifier responses [Falotico and Quatto, 2014, Grant et al., 2016]. This is easy to see in the limiting case where two classifiers have perfect accuracies ($p_1 = p_2 = 1$): Division by zero renders EC undefined because, intuitively, error consistency can not be evaluated in the absence of errors. Moreover, if exactly one classifier has perfect accuracy, EC will be 0 *irrespective of the accuracy of the other classifier*. This phenomenon, which we prove in Appendix C, may be counterintuitive and can lead to estimation instabilities, see below.

EC is bounded by the mismatch of the two classifiers' accuracies [Geirhos et al., 2020]. We provide an instructive visualization of these asymmetric bounds in Figure 2a. Only when the accuracies of both classifiers are equal can all values in $[-1, 1]$ be achieved. This implies that EC can never be high between two classifiers if their accuracies are substantially different. In practice, one can therefore not take EC at face value: EC has to be interpreted with the accuracies of the two classifiers in mind, and is, unfortunately, strongly susceptible to both floor- and ceiling-effects. The precondition for a straightforward interpretation of EC is thus a good match of classifiers' accuracies.

However, previous work has evaluated EC in conditions in which accuracies of the two classifiers were vastly different, see Figure 2b. Even without considering their joint probabilities, EC between classifiers must be low in these cases because their accuracy mismatch imposes an upper bound on the observable EC values.

To formalize this dependence of EC on classifiers' accuracy match, we present two propositions. First, when the accuracies of the two classifiers are equal, EC can best be understood as the proportion of copied responses: In a model in which the second classifier copies the responses of the first with probability $p_{copy}$ and responds independently at random otherwise, EC is related to that copy probability. In Proposition 1, we present this result as a general property of Cohen's $\kappa$ (for relating $\hat{y}^{(1)}$ and $\hat{y}^{(2)}$) which immediately translates to the application as EC (relating $r^{(1)}$ and $r^{(2)}$), where equal marginal distributions correspond to equal accuracies.

**Proposition 1.** *Let a classifier give responses according to some marginal distribution $\hat{y}^{(1)} \sim \tilde{P}^{(1)}$. If another classifier copies that response $\hat{y}^{(2)} = \hat{y}^{(1)}$ with probability $p_{copy}$ and otherwise draws independently from a marginal distribution $\hat{y}^{(2)} \sim \tilde{P}^{(2)}$ that is the same as the first $\tilde{P}^{(1)} = \tilde{P}^{(2)}$, then*

$$Cohen's \ \kappa = p_{copy}.$$

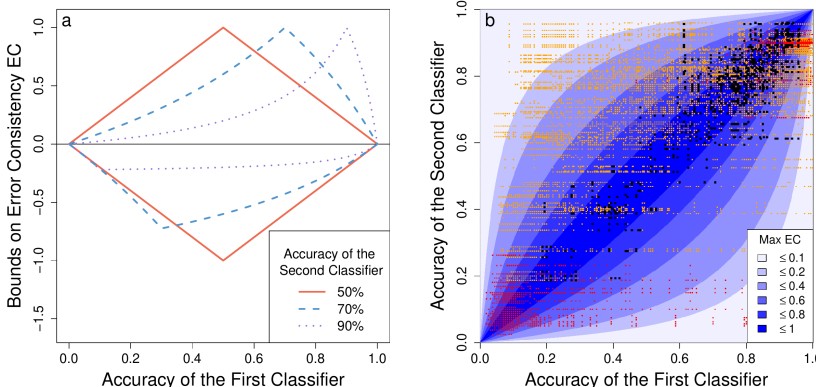

Figure 2: **Theoretical issues with EC.** (a) The bounds on error consistency (EC) depend on the mismatch between the accuracies of the two compared classifiers. (b) Similar to (a) but with accuracies on the x and y axes. Here, each orange dot corresponds to one Model-vs-Human comparison and each black dot to one human-vs-human comparison in the analysis of Geirhos et al. [2021]. Red dots correspond to conditions that were sampled but excluded for the analysis by the original authors. Human-vs-human comparisons have lower accuracy mismatches, so EC values are expected to be higher a priori.

*Proof.* The underlying distributions translate into the observed distributions $P^{(1)} = \tilde{P}^{(1)}$ and $P^{(2)} = p_{copy}\tilde{P}^{(1)} + (1 - p_{copy})\tilde{P}^{(2)}$. With $\tilde{P}^{(1)} = \tilde{P}^{(2)}$ we have $P^{(1)} = P^{(2)} = \tilde{P}^{(1)} = \tilde{P}^{(2)}$.

For the joint distribution, we have $P^{(1,2)}(y, y) = \tilde{P}^{(1)}(y)(p_{copy} + (1 - p_{copy})\tilde{P}^{(2)}(y))$ because the second classifier makes the same prediction as the first not only in all copy cases ($p_{copy}$), but also in those cases in which it does not copy but coincidentally makes the same prediction ($(1 - p_{copy})P^{(2)}(y)$). With that, $\kappa$ simplifies to $p_{copy}$.

$$\kappa = \frac{p_{obs} - p_{exp}}{1 - p_{exp}}$$

$$= \frac{\sum_{y=1}^{K} P^{(1,2)}(y, y) - \sum_{y=1}^{K} P^{(1)}(y)P^{(2)}(y)}{1 - \sum_{y=1}^{K} P^{(1)}(y)P^{(2)}(y)}$$

$$= \frac{\sum_{y=1}^{K} \left( P^{(1)}(y)(p_{copy} + (1 - p_{copy})\tilde{P}^2(y)) - P^{(1)}(y)P^{(2)}(y) \right)}{1 - \sum_{y=1}^{K} P^{(1)}(y)P^{(2)}(y)}$$

$$= \frac{\sum_{y=1}^{K} \left( P^{(1)}(y)(p_{copy} + (1 - p_{copy})P^{(1)}(y)) - P^{(1)}(y)P^{(2)}(y) \right)}{1 - \sum_{y=1}^{K} P^{(1)}(y)P^{(2)}(y)}$$

$$= \frac{\sum_{y=1}^{K} \left( P^{(1)}(y)p_{copy} + P^{(1)}(y)P^{(1)}(y) - P^{(1)}(y)P^{(1)}(y)p_{copy} - P^{(1)}(y)P^{(1)}(y) \right)}{1 - \sum_{y=1}^{K} P^{(1)}(y)P^{(1)}(y)}$$

$$= p_{copy} \frac{\sum_{y=1}^{K} P^{(1)}(y) - \sum_{y=1}^{K} P^{(1)}(y)P^{(1)}(y)}{1 - \sum_{y=1}^{K} P^{(1)}(y)P^{(1)}(y)}$$

$$= p_{copy} \frac{1 - \sum_{y=1}^{K} P^{(1)}(y)^2}{1 - \sum_{y=1}^{K} P^{(1)}(y)^2}$$

$$= p_{copy}$$

$\square$

Second, in the general case when the two classifiers have different accuracies, EC measures two aspects of similarity between the two classifiers:

(1) the proportion of copied responses ($p_{copy}$) and

(2) the (mis-)match between their accuracies (factor $f$ in Proposition 2).

Both are intuitively valid determinants of the similarity between two classifiers. We formalize the interplay between these two aspects in Proposition 2. In contrast, Safak [2020] tackled the same problem but proposed a linear scaling whereas our result reflects a more interpretable multiplicative scaling. Again, this result holds for Cohen's $\kappa$ in general and, therefore, also when it is applied to measure EC (for which $y$ would be replaced by $r$ and different marginals correspond to different accuracies).

**Proposition 2.** *Let a classifier give correct responses according to some marginal distribution $\hat{y}^{(1)} \sim \tilde{P}^{(1)}$. If another classifier copies that response $\hat{y}^{(2)} = \hat{y}^{(1)}$ with probability $p_{copy}$ and otherwise draws independently from its (potentially different) marginal distribution $\hat{y}^{(2)} \sim \tilde{P}^{(2)}$, then*

$$Cohen's\ \kappa = p_{copy} \cdot \underbrace{\frac{1 - \sum_y P^{(1)}(y)^2}{1 - \sum_y P^{(1)}(y)P^{(2)}(y)}}_{factor\ f\ due\ to\ mismatching\ marginals}$$

*with observed marginal distributions $P^{(1)} = \tilde{P}^{(1)}$ and $P^{(2)} = p_{copy}\tilde{P}^{(1)} + (1 - p_{copy})\tilde{P}^{(2)}$.*

See Appendix B for the proof. Crucially, this view of Cohen's $\kappa$ holds with generality as long as $K = 2$, without limiting assumptions about the marginal distributions. Note that the relation between EC and $p_{copy}$ is not symmetric: Whether classifier A is considered to copy from classifier B or vice versa results in different mismatch factors $f$. But note also that this factor vanishes ($f = 1$) if the reference classifier, from which responses are assumed to be copied, has a uniform marginal distribution.

This inspires a strong recommendation for applications of EC: When comparing multiple DNNs with different accuracies to humans, ensure that the *human responses* have uniform marginal distributions (i.e., have equally many correct as incorrect responses). In these cases, EC becomes interpretable as the proportion of responses that the DNNs copied from the human responses, EC $= p_{copy}$, regardless of the overall accuracy of the different DNNs. Without a uniform marginal distribution, DNNs with a higher-than-human accuracy may be evaluated as less similar to humans because their classification process is more accurate, not because it is functionally different. To be clear, we still believe that it is meaningful to interpret differences in accuracies between classifiers. But, ideally, EC would offer additional, orthogonal information beyond that accuracy mismatch—which is ensured by keeping the marginal distribution of the reference classifier (humans) uniform.

Beyond this contribution to interpreting EC, the generative model in Proposition 2 contributes in another way: It allows simulating data to predict the stability of real data. We will heavily use the latter to give guidance to practitioners planning how many samples they should present to real human participants in their experiments.

## 3 Calculating confidence intervals for EC values

Whenever an aggregate measure of empirical data is reported, it is best practice to also quantify the degree of uncertainty about the final value. Historically, this has not always been done for EC, e.g. none of Geirhos et al. [2018], Ollikka et al. [2024], Li et al. [2025] report confidence intervals. While [Geirhos et al., 2020] provide a basic variant of confidence intervals, these were computed in a suboptimal way: They were based on the expected agreement $p_{exp}$ rather than the actual accuracies of the two classifiers, $p_1$ and $p_2$. This creates an ambiguity because, for example, ($p_1 = 0.5, p_2 = 0.5$) as well as ($p_2 = 0.5, p_1 = 0.99$) both produce the same expected agreement of $p_{exp} = 0.5$ but result in different confidence interval widths. Moreover, these confidence intervals were necessarily centered around 0, because they were constructed only for independent observers. Our approach enables calculating confidence intervals for two dependent observers by making use of the observed accuracies. We begin by outlining how confidence intervals can be obtained for existing empirical data, using a straight-forward bootstrapping approach. After that, we make use of the generative model implied by Proposition 2 to derive confidence intervals ahead of time to plan sufficiently powerful experiments.

## 3.1 Using existing data for bootstrapping

Assuming that we already have access to two sequences of binary responses, $r^{(1)}$ and $r^{(2)}$ of length $N$, we can obtain a measure of uncertainty simply by *bootstrapping* [Efron and Tibshirani, 1994]. To do so, we sample $N$ elements with uniform probability and replacement from the sequences, $((r_i^{(1)}, r_i^{(2)}) \mid i \in \{1, ..., N\})$, and re-calculate EC based on the sampled trials. This procedure is established in other fields using consistency measures [McKenzie et al., 1996, Vanbelle and Albert, 2008]. Doing this repeatedly yields an empirical distribution of EC values, for which we can report a 95% confidence interval by taking the central 95% of the posterior mass, i.e. the interval $[q_{0.025}, q_{0.975}]$, where $q_p$ denotes the $p$-th quantile of the posterior distribution. Alternatively, one could report the Highest Posterior Density Interval (HDPI), which is the narrowest interval containing a certain amount (e.g. 95%) of the posterior mass. The only parameter of this procedure is the number of bootstraps, $M$, which we typically set to $M = 10,000$, see Figure 8 for a justification of this choice. For readers unfamiliar with the bootstrap approach, we provide a more detailed description in Appendix F and an explicit algorithm in Algorithm 1.

We apply this method of estimating CIs via bootstrapping to the data from Geirhos et al. [2021] in Section 4. In their work, several human observers (typically four to six) and 52 models were evaluated on corrupted natural images, which had to be classified into sixteen classes. There are 17 different corruptions, most of which are parameterized by a scalar intensity level, leading to multiple conditions within each corruption. Each condition contains 160 to 640 images. For every model, its EC to human observers is calculated in a hierarchical fashion: First, within each condition, the ECs to all human observers are calculated and then averaged. Then, the conditions of each corruption are averaged again, before calculating the final "human-machine error consistency" as an average over corruptions. To accurately reflect this procedure, we bootstrap through the entire calculation, obtaining a posterior over the final average itself and plot 95% percentile intervals of these averages in Figure 4.

## 3.2 Using Proposition 2 for simulations

But what about the scenario in which no data is available yet, e.g. when planning an experiment? Without data, we first need a stochastic generative model that can produce trial sequences resulting in a target EC, given two classifiers characterized by their marginals, i.e. their accuracies.

Proposition 2 provides such a model in which the underlying marginals and copy probability can be specified to generate data. Note that, depending on the copy probability, the observed marginals will deviate from the underlying marginal: The more responses the second classifier copies from the first, the more its marginal distribution adapts. To simulate data with specific marginal distributions $P^{(1)}$ and $P^{(2)}$ as well as EC, one can simulate data with copy probability and underlying marginals

$$p_{copy} = \text{EC} \cdot \left( \frac{1 - \sum_{y=1}^{K} P^{(1)}(y) P^{(2)}(y)}{1 - \sum_{y=1}^{K} P^{(1)}(y)^2} \right),$$

$$\tilde{P}^{(1)}(y) = P^{(1)}(y), \text{ and}$$

$$\tilde{P}^{(2)}(y) = \frac{P^{(2)}(y) - p_{copy} \cdot P^{(1)}(y)}{1 - p_{copy}}.$$

Equipped with this generative model, we can estimate confidence intervals for a pair of classifiers as follows: We first sample $N$ trials for the first classifier by drawing from a Bernoulli distribution $r_i^{(1)} \sim \tilde{P}^{(1)}$. Next, we copy $p_{copy} \cdot N$ responses from the first classifier to the second. For the remaining $(1 - p_{copy}) \cdot N$ responses, we sample independently from a Bernoulli distribution from the underlying marginal $r_i^{(2)} \sim \tilde{P}^{(2)}$. This yields one data set based on which we estimate EC. Repeating this process then yields an estimate for the variability of these EC values. We provide an algorithmic description of this approach in Algorithm 2.

We plot the resulting confidence intervals in Figure 3, and include sample sizes used in the literature [Geirhos et al., 2021, Ollikka et al., 2024, Li et al., 2025, Wiles et al., 2024] as reference points. We focus on the most severe case where $\kappa = 0.5$, but discuss the effect of $\kappa$ on CI size in Appendix E. Note that the x-axis is on a logarithmic scale and reaches values that are very difficult to meet in experiments with human observers. Even so, the width of the confidence intervals (especially that

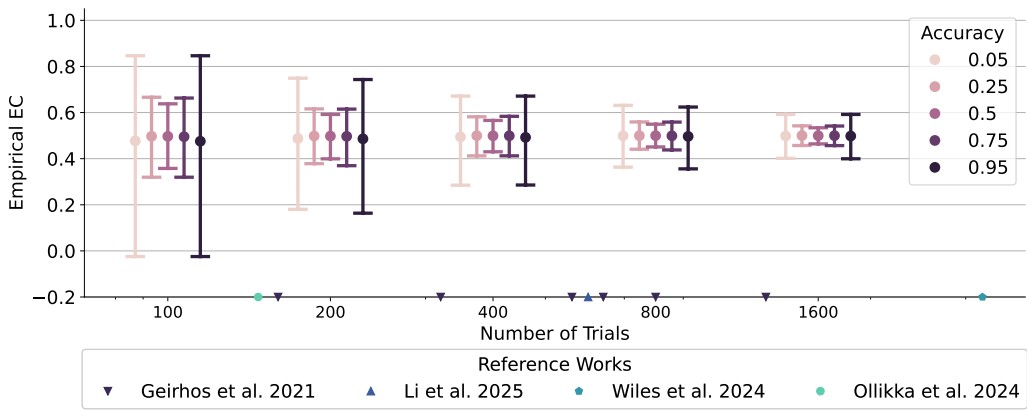

Figure 3: **95% Confidence Interval sizes as a function of trial number.** We simulate data demonstrating how the size of EC confidence intervals depends on both the accuracy of the classifiers and the number of trials. The ground truth EC is 0.5, and we bootstrap 1,000 times for each pair of observers, which are set to the same accuracy for simplicity. We also include the trial numbers employed in the literature for reference (symbols on the x-axis).

of classifiers close to floor / ceiling performance) is very large and shrinks only slowly, covering 0.2 units of $\kappa$ even at $N = 400$ trials, which is roughly the size of the difference found between humans and machines by Geirhos et al. [2021]. Especially noteworthy is the fact that at lower $N$ and performance close to floor / ceiling, the bootstrapped ECs are biased and undershoot the true value. This happens for a simple reason: If one classifier is always correct (or always wrong) the EC will be zero irrespective of the accuracy of the other one. In the critical performance regimes and at a low number of trials, sub-sampling trial sequences for which these edge cases occur becomes likely enough to bias results.

**Limitations.** A core limitation of our work is that we do not present an analytic solution for confidence intervals, but rely on bootstrapping instead. This also limits the numerical accuracy of the significance tests, which cannot resolve p-value differences $\leq \frac{1}{N}$. While the analytic solutions (along the lines of Gwet [2016], Fleiss et al. [1969], and Vanbelle et al. [2024]) could improve this slightly, we believe that for the discussed purpose here, our methodology is the pragmatic choice that comes with fewer restrictions (and is certainly much better than not reporting confidence intervals at all).

### 3.3 Significance Tests for Error Consistency

Now that we can quantify the degree of uncertainty, we can also derive significance tests for error consistency. Significance tests are an invaluable tool for empirical scientists because they provide a standardized way of checking if results are statistically reliable. Given two sequences of binary trials of length $N$ which lead to an EC of $\kappa$, we want to calculate a p-value for $\kappa$. The p-value is the probability of observing an event at least as extreme as the observed one under the null hypothesis of two independent binomial observers, i.e. $\kappa = 0$.

**Basic Significance Test.** To obtain the distribution of EC values obtained under the null hypothesis $H_0 : \kappa = 0$, we first need to model the independent binomial observers that constitute the null hypothesis. They are fully characterized by their individual accuracies, of which the empirically observed accuracies provide an estimate. The reliability of this estimate depends on the number of trials. If we assume uniform priors over these accuracies, the posteriors are beta-distributions characterized by the number of (in)correctly solved trials, $\text{Beta}(N - \sum_i r_i^{(j)}, \sum_i r_i^{(j)})$.

We can now sample the two individual accuracies from their respective posteriors, sample $N$ trials from independent binomial distributions and calculate the resulting EC. Repeating this procedure $M$ times leads to a distribution of EC values expected under the null hypothesis. From this distribution,

we then calculate how many times the absolute value of the simulated EC exceeded the absolute value of the empirically measured EC, for a two-sided significance test. We demonstrate this method by applying it to data from Model-vs-Human in Appendix H to obtain p-values of the error consistency between a human subject and the best-performing model, a CLIP-trained ViT [Radford et al., 2021].

**Advanced Significance Test.**    The empirical scenario which we investigate in Section 4 demands a slightly different significance test: Given a fixed reference observer (e.g. a human) and two candidate observers (e.g. two DNNs), is the difference of ECs to the reference observer found between the two candidates statistically significant?

Such a test is also possible. Let $\kappa_{c_1}, \kappa_{c_2}$ be the error consistencies of the two candidates. We consider the null hypothesis of the candidates having the same error consistency to the reference classifier, $H_0 : \kappa_{c_1} = \kappa_{c_2}$. We first obtain a distribution of error consistency differences via bootstrapping as above: First, we randomly sample $N$ trial indices $i$ with replacement and uniform probability. We select the corresponding response-correctness-triplets $(r_i^{(ref)}, r_i^{(c_1)}, r_i^{(c_2)}) \in \{0, 1\}^3$ and re-calculate EC values and their difference. To obtain the distribution of differences one would expect under $H_0$, we simply exchange $r_i^{(c_1)}, r_i^{(c_2)}$ with probability $p = 0.5$, as if there had been only one underlying model. We can then perform a t-test between the distributions of differences to compute p-values.

# 4    Revisiting earlier work

## 4.1    Model-vs-Human

Next, we re-analyze well-known literature results by calculating confidence intervals around their empirically measured values. At the focus of our attention is the work by Geirhos et al. [2021], where a cohort of human observers was asked to classify corrupted natural images. The same images were also shown to 52 DNNs, to evaluate their alignment to human behavior. EC values were calculated per condition and averaged as described in Section 3. From our analysis in Figure 3, we already know that some of the error consistencies were calculated in a low-data regime that will result in large confidence intervals around the individual EC values, but we now compute confidence intervals specifically for their empirical data. We do this by bootstrapping through the entire calculation, to arrive at an empirical distribution of final scores, which we plot for humans and every model in Figure 4. The best model is a CLIP-trained ViT[Radford et al., 2021] which performs much better on the distribution shifts like sketches and stylized images, achieving an EC to humans of $0.279$. The second-best model, BiTM-ResnetV2-101x1, achieves an EC of $0.251$, resulting in a difference of $0.028$, which, as a two-sided t-test confirms, is significantly different from 0 ($p < 1 \times 10^{-10}$). But note how model 24 (VGG-19 as per Table 2) has overlapping CIs with all models from 8 to 45, more than 70% of all models. It is conceivable that if these experiments were done again, VGG-19 might jump up to rank 8, or down to rank 45. It just so happens that the very best and very worst models broaden the range we have to plot, thus rendering this effect less obvious than in Figure 1. Testing the differences for all neighboring models for significance, we find no other significant result, even without correcting for multiple comparisons.

For each bootstrap, we also obtain a ranking of models. Given that the CIs are fairly wide (for example, no model's confidence interval is separated from that of its neighbors), we wonder whether different bootstraps imply different model rankings. To check this, we calculate Kendall's $\tau$ between the original model ranking and the ranking implied by each of the 10k bootstraps. $\tau$ is a statistical measure of the rank correlation between two ordered sequences, similar to Spearman's rank correlation, but with the benefit of an intuitive interpretation: $\tau$ measures the difference between the probability that a pair of observations is concordant and the probability that it is discordant. The average $\tau$ is $0.88$, which means that two randomly sampled pairs have an 88% chance of being concordant. Evidently, the main conclusion drawn by Geirhos et al. [2021] clearly still holds: Humans are more consistent among themselves than models are to humans. Nevertheless, adding confidence intervals to the reported EC values better quantifies the degree of uncertainty about the measurements for individual models, and the 95% CIs do indeed overlap, meaning that differences between models are not resolved.

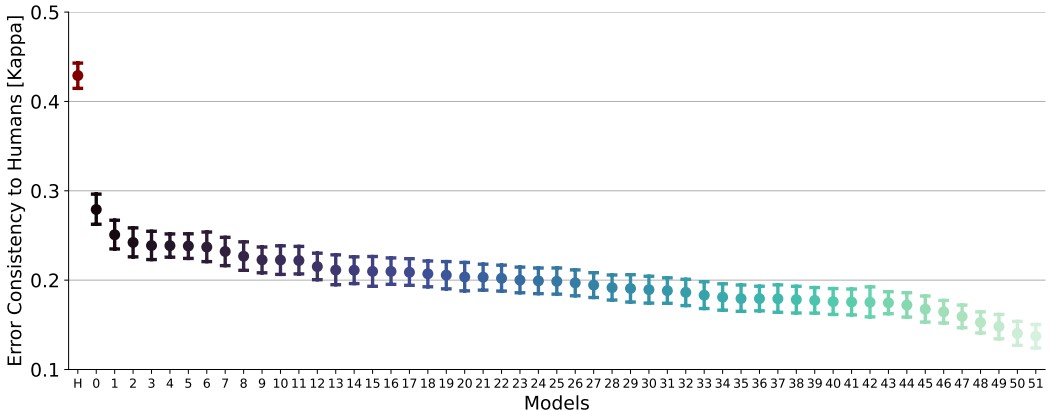

Figure 4: **Models from MvH ranked by their EC to humans.** Inter-human EC is plotted in red. We plot the 95% percentile interval for the mean EC obtained via bootstrapping, using the standard exclusion criteria of the benchmark (orange and black points in Figure 2b). For the mapping of model indices to names, see Table 2. Note that while the gap to humans is large, confidence intervals of many models overlap.

## 4.2 Brain-Score

One of the most widely-known applications of EC is the influential Brain-Score benchmark[2] [Schrimpf et al., 2018], which evaluates arbitrary DNNs by how much they resemble the brain. One dimension of this evaluation is the behavioral similarity to humans, including error consistency as measured via the data by Geirhos et al. [2021]. Here we assess how reliable the ranking of models implied by their EC to humans really is.

In Brain-Score, the EC for each condition is first divided by the human-to-human EC for this condition, to express values relative to a ceiling. We begin by reverting this operation to obtain raw EC values. To properly bootstrap EC values for these models, we would need access to the complete sequences of trials generated by the models, but this data is not stored anywhere, and re-running the models (not all of which are publicly available anyway) seems excessive. Hence, we instead make mild assumptions to estimate conservative CIs around the empirical values, demonstrating the utility of our copy-model. (See Appendix I for details on this procedure.) We obtain the result depicted in Figure 5 for the top-30 models (by EC to humans). Again, the best model is a CLIP-trained ViT. For the other models, the CIs are so large that any of the top ten models could be the second-best one, meaning that the true rank order between models remains unclear.

We believe that the issue we raise here generalizes beyond just EC and Brain-Score: Any metric that is used to derive rankings of models in a benchmark should come with a method for calculating confidence intervals. Likewise, any benchmark that provides a ranking should quantify the degree of uncertainty surrounding individual values and the stability of the resulting ranking.

## 5 Discussion

In this work, we have built on the error consistency metric proposed by Geirhos et al. [2020], demonstrating how confidence intervals and significance tests for empirical measurements of EC can be calculated using bootstrapping techniques. Additionally, we have presented a new computational model of EC for planning sufficiently powerful, conclusive experiments. We have used our methodology to revisit influential Neuro-AI results Schrimpf et al. [2018], finding that many results are not statistically reliable, because previous studies were underpowered. In the following, we recommend best practices for future benchmarking of behavioral alignment, and facilitate their implementation by making our python package publicly available. A core contribution of our work is an improved understanding of error consistency through a new computational model: EC is broken down into

---

[2] https://www.brain-score.org

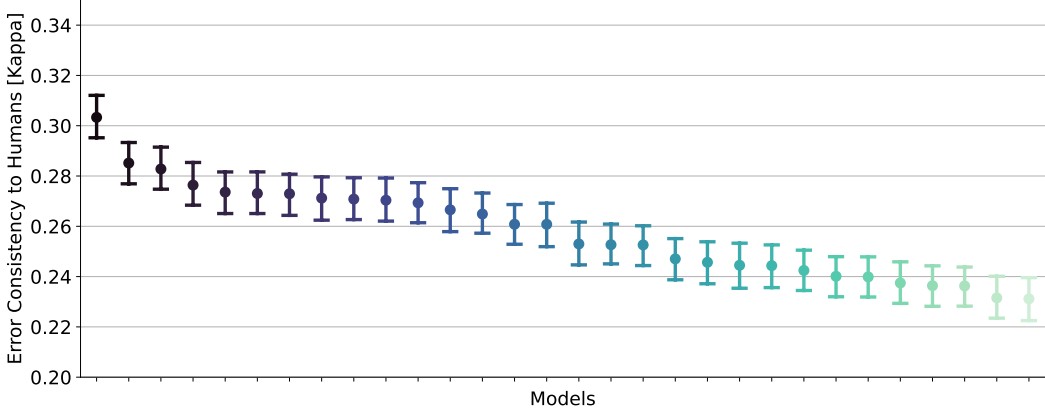

Figure 5: **Confidence Intervals for Brain-Score ECs.** Like in Figure 4, we plot the 95% confidence interval of the EC of the top 30 Brain-Score models. Note that the confidence intervals of many mid-range models overlap.

two interpretable components, copy probability $p_{copy}$ and marginal mismatch $f$. As an analogy, this is similar to breaking down accuracy into true positive and false positive rates—a necessary shift in perspective that will improve interpretations in future studies measuring similarities between classifiers.

## 5.1  Practical Recommendations: How to reliably benchmark behavioral alignment

From our analysis, we can derive concrete recommendations for practitioners. As shown in Figure 3, a sufficiently high number of trials is required to obtain small confidence intervals around EC values. As a rule of thumb, we suggest to collect at least 1000 trials per classifier to balance this requirement against practical constraints. In experiments with human observers, achieving very high resolution of EC differences could become prohibitively expensive, underlining the need for a good selection of stimuli, which probe differences efficiently. Since the size of CIs also depends on the accuracy mismatch between classifiers, we suggest to aim for an accuracy around 75%, but not more than 90% to avoid ceiling-effects, which can be severe. The same holds for floor-effects. This is in agreement with standard best practices from psychophysics, where one aims for a similar accuracy level in human observers to sustain motivation whilst avoiding ceiling effects. Finally, future work should always report confidence intervals and check results for statistical significance, which can easily be done with the python package we provide.

## 5.2  Conclusion

The field of ML research relies heavily on the powerful machinery of benchmarking [Hardt, 2025]. Hence, the correctness and reliability of benchmarks is crucial for progress, since ill-calibrated benchmarks will lead us astray. The field is also notorious for not quantifying uncertainty properly [Pineau et al., 2021, Miller, 2024, Lehmann and Paromau, 2025, Bouthillier et al., 2021], which is especially important in the context of benchmarks because of their large influence on other developments. Our work raises general questions about how benchmarks should aggregate results. For benchmarks that explicitly compute rankings, we give the concrete recommendations of (a) providing confidence intervals around individual estimates and (b) quantifying the stability of the resulting rankings as we did with error consistency.

In this work, we have revisited results by Geirhos et al. [2021] which influence the Brain-Score benchmark Schrimpf et al. [2018], revealing that many differences reported between deep vision models are not statistically reliable. The error consistency metric is particularly susceptible to such issues, because it is inherently noisy, so the requirements for stable measurements are hard to meet in practice. We envision that our work will help the field to overcome these issues, by providing tools to plan sufficiently powerful experiments.

## Acknowledgments and Disclosure of Funding

Funded, in part, by the Collaborative Research Centre (CRC) "Robust Vision – Inference Principles and Neural Mechanisms" of the German Research Foundation (DFG; SFB 1233), project number 276693517. FAW acknowledges funding by the BBVA Foundation Programme Grant "Harnessing Vision Science to Overcome the Critical Limitations of Artificial Neural Networks". This work was additionally supported by the German Federal Ministry of Education and Research (BMBF): Tübingen AI Center, FKZ: 01IS18039A. WB acknowledges financial support via an Emmy Noether Grant funded by the German Research Foundation (DFG) under grant no. BR 6382/1-1 and via the Open Philanthropy Foundation funded by the Good Ventures Foundation. WB, FAW and KM are members of the Machine Learning Cluster of Excellence, EXC number 2064/1 – Project number 390727645. KM is supported by the Carl Zeiss Foundation through the CZS Institute for Artificial Intelligence and Law. This research utilized compute resources at the Tübingen Machine Learning Cloud, DFG FKZ INST 37/1057-1 FUGG. The authors thank the International Max Planck Research School for Intelligent Systems (IMPRS-IS) for supporting Thomas Klein.

We would like to thank Robert Geirhos for helpful discussions, and for providing us with his raw data for Model-vs-Human. We also thank all anonymous reviewers, who provided valuable feedback during the rebuttal period, which improved the final manuscript.

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

# A Related Work

**DNNs as vision models.** Within the last decade, Deep Neural Networks (DNNs) have been proposed as models of the human visual system [Doerig et al., 2023, Kriegeskorte, 2015, Cichy and Kaiser, 2019, Kietzmann et al., 2017], not only because they form the only class of image-computable models that achieves human-level performance on benchmark tasks like ImageNet [Russakovsky et al., 2015], but also because performance-optimized DNNs are the best predictors of biological neural activations [Yamins et al., 2014, Kubilius et al., 2019, Zhuang et al., 2021]. However, the suitability of DNNs as models of vision is debated [Bowers et al., 2022, Wichmann and Geirhos, 2023, Maniquet et al., 2024, Serre, 2019], as is the question of how comparisons should best be conducted [Firestone, 2020, Lonnqvist et al., 2021, Wichmann and Geirhos, 2023].

**Error Consistency.** The underlying idea motivating error consistency is that of "molecular psychophysics" [Green, 1963], which is to go beyond aggregate measures like accuracy, and instead compare behavior on a trial-by-trial basis. EC promises to achieve this by building on Cohen's $\kappa$ [Cohen, 1960, Martín Andrés and Álvarez Hernández, 2024], which considers the trial-level agreement observed for a pair of classifiers. Geirhos et al. [2020] proposed EC in the context of human-machine comparison, but in principle, the method applies to the general setting of comparing arbitrary classifiers. Martín Andrés and Álvarez Hernández [2024] observe that $\kappa$ is a biased estimator and should be corrected, which we support in our implementation. Safak [2020] also observe the problem of $\kappa's$ dependency on the marginals, which we outline in Section 2. They propose to scale $\kappa$ between its theoretical limits in an attempt to obtain a measure of error consistency that is orthogonal to accuracy. Goel et al. [2025] propose to apply Cohen's $\kappa$ directly to the responses given by both classifiers $\hat{y}_i$ rather than the correctness values $r_i$, and adapt the marginal probabilities of independent classifiers to reflect their accuracy. We still focus on the standard error consistency proposed by Geirhos et al. [2020] because it has seen the widest application. In principle, our bootstrapping methodology also works for these other metrics.

**Uncertainties in Benchmarking.** Benchmarking has a long history and is central to modern machine learning [Koch and Peterson, 2024, Singh et al., 2025, Hardt, 2025, Orr and Kang, 2024]. For example, the popular Papers with Code database[3] lists approximately 14,000 different benchmarks. With the emergence of modern benchmarks and leaderboards, issues such as overfitting to benchmarks [Singh et al., 2025], the limitations of metrics [Thomas and Uminsky, 2020], and various sources of uncertainty have been identified and challenged [Lehmann and Paromau, 2025, Bouthillier et al., 2021]. In general, it has been criticized that benchmarks should not be the primary goal of scientific machine learning research [Alzahrani et al., 2024]. At the same time, it has been pointed out that benchmarking itself requires a scientific approach and some kind of standard methodology [Thiyagalingam et al., 2022, Sculley et al., 2018, Hardt, 2025]. Similar to our work, Nado et al. [2021] argued that uncertainty is an important aspect to consider for benchmarks. In contrast to previous work, we focus on uncertainty in the behavioral comparison of classifiers using error consistency.

# B Proof of Proposition 2

*Proof.* The underlying distributions translate into the observed distributions $P_1 = \tilde{P}_1$ and $P^{(2)} = p_{copy}\tilde{P}^{(1)} + (1 - p_{copy})\tilde{P}^{(2)}$. Inverting the marginal $P^{(2)}$, we get $\tilde{P}^{(2)}(y) = \frac{P^{(2)}(y) - p_{copy} \cdot P^{(1)}(y)}{1 - p_{copy}}$.

For the joint distribution, we have $P(y, y) = \tilde{P}^{(1)}(y)(p_{copy} + (1 - p_{copy})\tilde{P}_2(y))$ because the second classifier makes the same prediction as the first in all copy cases ($p_{copy}$) and in those cases, in which it does not copy but coincidentally makes the same prediction ($(1 - p_{copy})P^{(2)}(y)$).

---

[3]https://paperswithcode.com/sota

$$\kappa = \frac{p_{obs} - p_{exp}}{1 - p_{exp}}$$

$$= \frac{\sum_{y=1}^{K} P^{(1,2)}(y,y) - \sum_{y=1}^{K} P^{(1)}(y)P^{(2)}(y)}{1 - \sum_{y=1}^{K} P^{(1)}(y)P^{(2)}(y)}$$

$$= \frac{\sum_{y=1}^{K} \left( P^{(1)}(y)(p_{copy} + (1 - p_{copy})\tilde{P}_2(y)) \right) - \sum_{y=1}^{K} P^{(1)}(y)P^{(2)}(y)}{1 - \sum_{y=1}^{K} P^{(1)}(y)P^{(2)}(y)}$$

$$= \frac{\sum_{y=1}^{K} \left( P^{(1)}(y)(p_{copy} + P^{(2)}(y) - p_{copy} \cdot P^{(1)}(y)) \right) - \sum_{y=1}^{K} P^{(1)}(y)P^{(2)}(y)}{1 - \sum_{y=1}^{K} P^{(1)}(y)P^{(2)}(y)}$$

$$= p_{copy} \frac{\sum_{y=1}^{K} \left( P^{(1)}(y) - P^{(1)}(y) \cdot P^{(1)}(y) \right)}{1 - \sum_{y=1}^{K} P^{(1)}(y)P^{(2)}(y)}$$

$$= p_{copy} \cdot \left( \frac{1 - \sum_{y=1}^{K} P^{(1)}(y)^2}{1 - \sum_{y=1}^{K} P^{(1)}(y)P^{(2)}(y)} \right)$$

$\square$

## C  Proof of Ceiling Effects

In Section 2, we claim that as soon as one classifier achieves perfect accuracy on the investigated sequence of trials, the EC of the pair will be 0 irrespective of the accuracy of the other one. Here, we proof this claim, which is easiest to derive from a confusion matrix as shown in Table 1. The observed agreement is found on the diagonal, $p_{obs} = a + d$, while the expected agreement is what would be on the diagonal if observers were independent ($p_{exp} = (a + b)(a + c) + (c + d)(b + d)$). For the proof, consider without loss of generality that classifier 2 was perfect, i.e. $a + c = 1$.

|  | $r^{(2)} = 1$ | $r^{(2)} = 0$ |  |
|---|---|---|---|
| $r^{(1)} = 1$ | $a$ | $b$ | $a + b$ |
| $r^{(1)} = 0$ | $c$ | $d$ | $c + d$ |
|  | $a + c$ | $b + d$ | 1.0 |

Table 1: Error consistency matrix.

*Proof.* Let $a + c = 1$, then $b + d = 0$, so $p_{exp} = (a + b)(a + c) + (c + d)(b + d) = (a + b)$. Since $b \geq 0$ and $d \geq 0$, $b = d = 0$.

$$\kappa = \frac{p_{obs} - p_{exp}}{1 - p_{exp}}$$

$$= \frac{(a + d) - (a + b)}{1 - (a + b)}$$

$$= \frac{d - b}{1 - (a + b)}$$

$$= 0$$

$\square$

By symmetry, the same argument holds for the case where one classifier gives incorrect responses on all trials.

## D  Lower bound on EC

Figure 2b shows how the mismatch in accuracies between two classifiers imposes upper bounds on EC. In extension, Figure 6 shows that there are also non-trivial lower bounds (subplot b). These lower bounds are not a simple function of the upper bounds. Instead, they are inversely related such that when accuracies are inverted (i.e., $p_1 = 1 - p_2$ along the off-diagonal) lower values can be attained. But perhaps counterintuitively, even with inverted accuracies, EC $= -1$ can not always be reached. This is in contrast to the upper bound, where aligned accuracies (i.e., $p_1 = p_2$) always allows EC $= +1$.

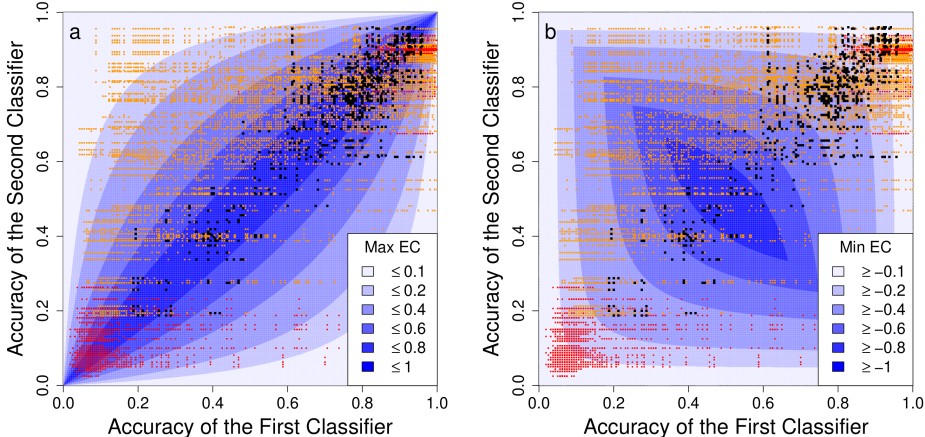

Figure 6: **Lower bound of EC in comparison to upper bounds.** The left subplot (a) is the same as 2b, showing regions in which EC is upper bounded by the mismatch in accuracies of two classifiers (x and y axes). Additionally, the right subplot (b) shows the lower bounds. Again, each point represents a Model-vs-Human comparison in the analysis of Geirhos et al. [2021].

## E  Effect of EC on CI size

For Figure 3, we have chosen a ground-truth EC of 0.5 to demonstrate how the EC between two classifiers depends on their accuracy and the number of trials on which EC is measured. However, the choice of the ground-truth EC influences the CI size as well, which we demonstrate in Figure 7. We again generate data using our copy-model, and plot the ground-truth EC (input of the model) against the distribution of empirical ECs (calculated on model outputs). To better show the changes in CI size, we plot the delta between empirical and ground-truth EC rather than the absolute EC. Note how the CI bounds are not symmetric and largest at a ground-truth EC of 0.5.

## F  Bootstrapping

For readers unfamiliar with bootstrapping, we provide a basic introduction here, but refer to Efron and Tibshirani [1994] for details. The idea is the following: We want to estimate the uncertainty in a summary statistic (e.g., the mean), which is computed for a dataset $X$ of size $N$. If we had access to the underlying data-generating process, we could simply generate $M$ more datasets (each of size $N$) and compute the summary statistic for each. This would give us a distribution of values of the statistic, from which we could then compute a measure of uncertainty (e.g. a percentile interval of the distribution of means). Without access to the true data-generating process, we build the best possible parameter-free model: To generate a data point, randomly select one of the data points $x_i$ with uniform probability. To generate an entire dataset, draw $N$ data points with uniform probability, with replacement. In the context of error consistency between two observers, one data point corresponds to one trial, i.e. a pair of responses $r_i = (r_i^{(1)}, r_i^{(2)}) \in \{0,1\}^2$. The full algorithm to obtain a confidence interval for two sequences of responses $r^{(1)}$ and $r^{(2)}$ can thus be summarized in Algorithm 1.

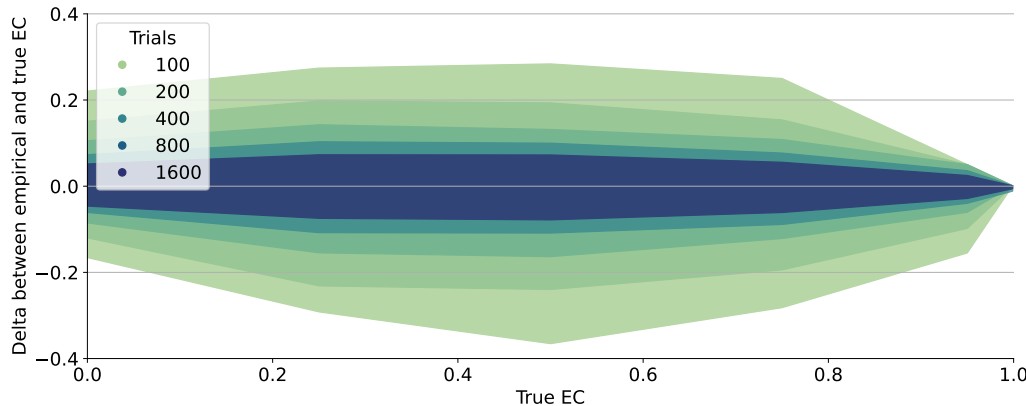

Figure 7: **Ground-Truth EC against CI sizes.** We simulate data with different ground-truth EC values and plot the size of 95% confidence intervals. Evidently, one obtains the largest CIs at ground-truth EC of 0.5.

---

**Algorithm 1** Bootstrapping EC confidence intervals

---

**Require:** Arrays $r^{(1)}, r^{(2)} \in \{0,1\}^N$, number of bootstrap iterations $M$, sample size $N$
**Ensure:** 95% confidence interval for error consistency
1: $\mathcal{L} \leftarrow \emptyset$          ▷ Initialize list of EC values
2: **for** $i = 1$ to $M$ **do**
3:     $\mathcal{I} \leftarrow$ sample $N$ indices from $\{1, \ldots, N\}$ uniformly with replacement
4:     $ec \leftarrow \text{EC}(r^{(1)}[\mathcal{I}], r^{(2)}[\mathcal{I}])$          ▷ Compute error consistency
5:     $\mathcal{L} \leftarrow \mathcal{L} \cup \{ec\}$          ▷ Add to list
6: **end for**
7: $[\text{lower}, \text{upper}] \leftarrow \text{Percentile}_{95\%}(\mathcal{L})$          ▷ Compute 95% interval
8: **return** $[\text{lower}, \text{upper}]$

---

Similarly, the algorithm for estimating confidence intervals without access to real trials is described in Algorithm 2.

# G    Validating the copy model

As proven by Proposition 2, our copy model is correct in the limit of trials. Additionally, we empirically validate the uncertainty estimates we derive from this model, by first generating some ground-truth data with a fixed error consistency at a fixed accuracy of both classifiers. We then bootstrap this data $M = 1,000$ times, as we would to obtain percentile intervals. Next, we instead draw $M$ samples from our copy model, to compare the CIs implied by our copy model to those implied by the bootstrap, which we consider the gold standard. We plot this comparison in Figure 9. Evidently, the distributions we obtain from our copy model match the bootstrapped results quite well, as long as the number of trials is sufficient or the accuracy avoids floor- and ceiling-effects.

# H    Details on Model-vs-Human results

An idiosyncrasy of the Model-vs-Human benchmark that we have not expanded on so far is the fact that while the final scores (which are averages across the different corruptions) are stable, the uncertainty within each corruption can be quite large. In Figure 11, we plot the CIs surrounding the EC values obtained for the phase-scrambling corruption as an example. In Figure 12 we plot the pairwise rank correlations between all 17 corruptions, revealing that some, but not all corruptions are highly correlated, i.e. imply similar model rankings. In Figure 10, we plot the confidence intervals one would obtain under the alternative aggregation strategy of simply averaging across conditions,

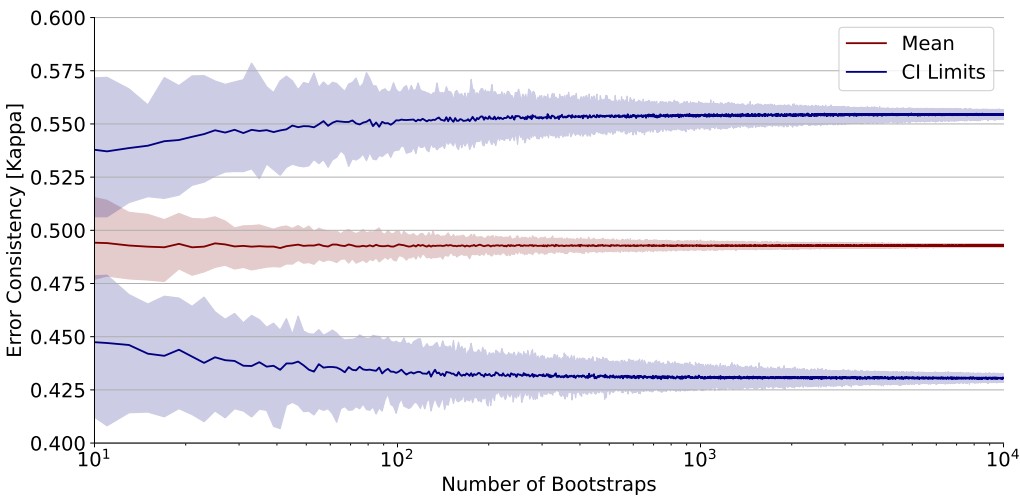

Figure 8: **Convergence of EC estimates as a function of number of bootstraps.** Starting from a single sequence of $1,000$ trials generated by our copy model (arbitrary target EC of $0.5$, classifier accuracies $0.7$ and $0.8$ respectively) we chose a number of bootstrap runs and bootstrapped 100 times for each, to obtain estimates of uncertainty. Shaded regions correspond to 95% percentile intervals. Evidently, at the $M = 10,000$ bootstrap runs we use, the estimates have stabilized, thus justifying our choice of $M$.

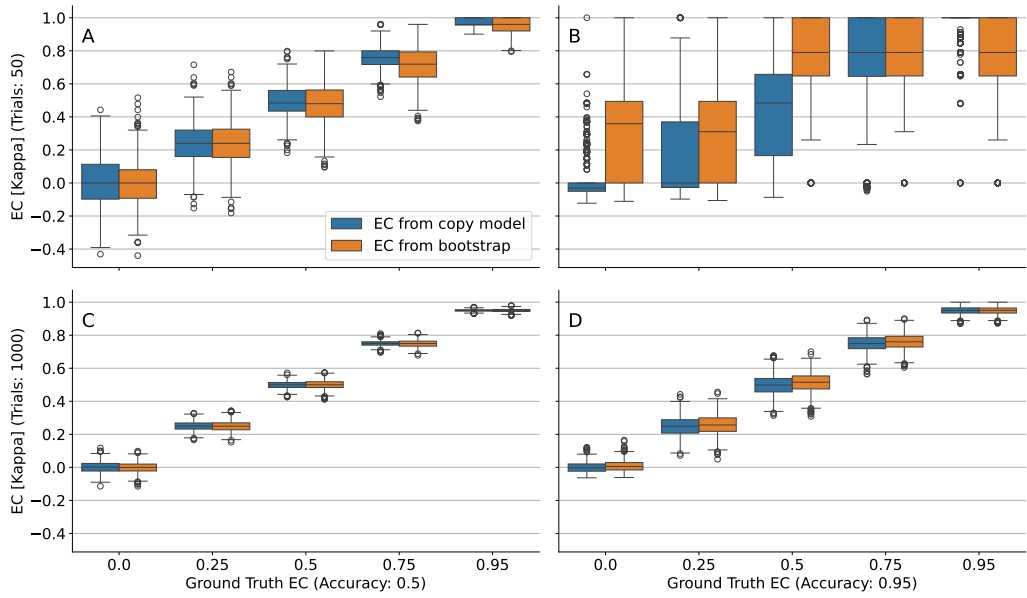

Figure 9: **Validating our copy model.** We plot the distributions of EC estimates from the copy model to bootstrapped (gold standard) estimates of uncertainty. **Top:** 50 trials. **Bottom:** 1000 trials. **Left:** Accuracy 50%. **Right:** Accuracy 95%. Evidently, if both the number of trials is low and accuracy approaches the ceiling of 100%, the model diverges from bootstrapped results, but error consistency is not well-behaved in this case anyway. Ideally, experiments should avoid this regime, which can be achieved by increasing the number of trials and calibrating their difficulty.

---

**Algorithm 2** Bootstrapping EC confidence intervals without real trials

---

**Require:** Target marginals $P^{(1)}, P^{(2)}$ over $K$ classes, target error consistency EC, number of trials $N$, number of simulations $M$

**Ensure:** $95\%$ confidence interval for error consistency

1: **Compute underlying parameters:**

2: $p_{\text{copy}} \leftarrow \text{EC} \cdot \left( \frac{1-\sum_{y=1}^{K} P^{(1)}(y)P^{(2)}(y)}{1-\sum_{y=1}^{K} P^{(1)}(y)^2} \right)$

3: $\tilde{P}^{(1)}(y) \leftarrow P^{(1)}(y)$ for all $y \in \{1, \ldots, K\}$

4: $\tilde{P}^{(2)}(y) \leftarrow \frac{P^{(2)}(y)-p_{\text{copy}} \cdot P^{(1)}(y)}{1-p_{\text{copy}}}$ for all $y \in \{1, \ldots, K\}$

5:

6: $\mathcal{L} \leftarrow \emptyset$          $\triangleright$ Initialize list of EC values

7: **for** $i = 1$ to $M$ **do**

8:      **Generate responses for first classifier:**

9:      **for** $j = 1$ to $N$ **do**

10:          $r_j^{(1)} \sim \text{Categorical}(\tilde{P}^{(1)})$          $\triangleright$ Sample from marginal

11:      **end for**

12:

13:      **Generate responses for second classifier:**

14:      $N_{\text{copy}} \leftarrow \lfloor p_{\text{copy}} \cdot N \rfloor$          $\triangleright$ Number of responses to copy

15:      $\mathcal{I}_{\text{copy}} \leftarrow$ sample $N_{\text{copy}}$ indices from $\{1, \ldots, N\}$ without replacement

16:      **for** $j = 1$ to $N$ **do**

17:          **if** $j \in \mathcal{I}_{\text{copy}}$ **then**

18:              $r_j^{(2)} \leftarrow r_j^{(1)}$          $\triangleright$ Copy response

19:          **else**

20:              $r_j^{(2)} \sim \text{Categorical}(\tilde{P}^{(2)})$          $\triangleright$ Sample independently

21:          **end if**

22:      **end for**

23:

24:      $ec \leftarrow \text{EC}(r^{(1)}, r^{(2)})$          $\triangleright$ Compute error consistency

25:      $\mathcal{L} \leftarrow \mathcal{L} \cup \{ec\}$

26: **end for**

27: $[\text{lower}, \text{upper}] \leftarrow \text{Percentile}_{95\%}(\mathcal{L})$          $\triangleright$ Compute $95\%$ interval

28: **return** $[\text{lower}, \text{upper}]$

---

without grouping by corruption first. Evidently, this would drastically affect the confidence intervals. To be clear, this is a deviation from the MvH protocol and thus "wrong", but the data itself is the same, thus illustrating the point that there is large variance in the data, which is obfuscated by the hierarchical aggregation.

In Figure 2b, we explain how in many of the conditions within Model-vs-Human, the accuracy mismatch between models and human participants is so large that $k_{max}$ takes on very small values. One might wonder how Model-vs-Human scores would change if one excluded conditions not as prescribed by the authors, but based on this mismatch. However, this would render the selection of conditions dependent on the model. Hence, the benchmark would no longer present a fair comparison of models, and final scores would no longer be comparable.

# I    Generating Synthetic Brain-Score Data

For the analysis conducted in Section 4.2, we estimate the size of confidence intervals surrounding models on the public Brain-Score benchmark. To estimate a model's EC to humans within one condition $c$, we would technically have to create one set of binary trials that has the property of having the desired average EC to the $n$ human observers of $\kappa_{real,c}$. To simplify, we drop this dependence and generate $n$ independent sequences of trials, one for each human who saw images in this condition. Generating only one sequence of trials would drastically overestimate the variance, which is reduced by taking the mean. For each of these sequences, we need three inputs for the copy-model: The

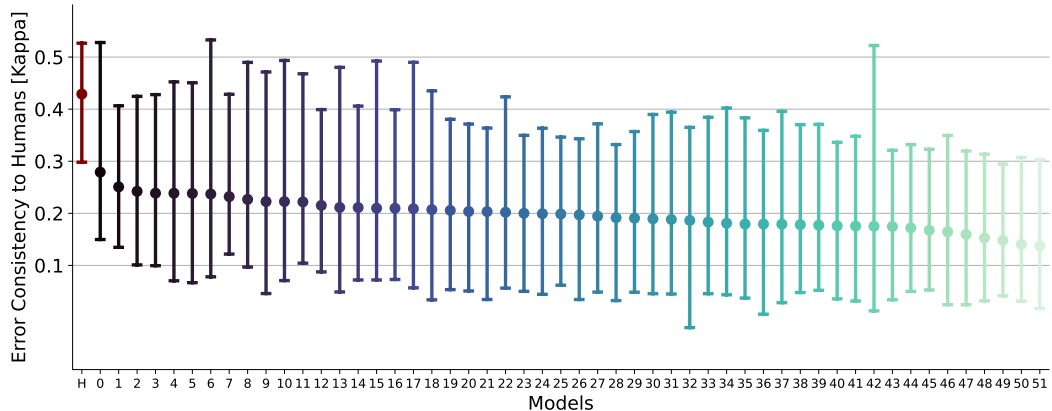

Figure 10: **EC to humans without hierarchical aggregation.** If one were to simply define a model's EC to humans as its average EC over all conditions (without first grouping by corruption), one would obtain the 95% confidence intervals depicted here, which clearly overlap for all models.

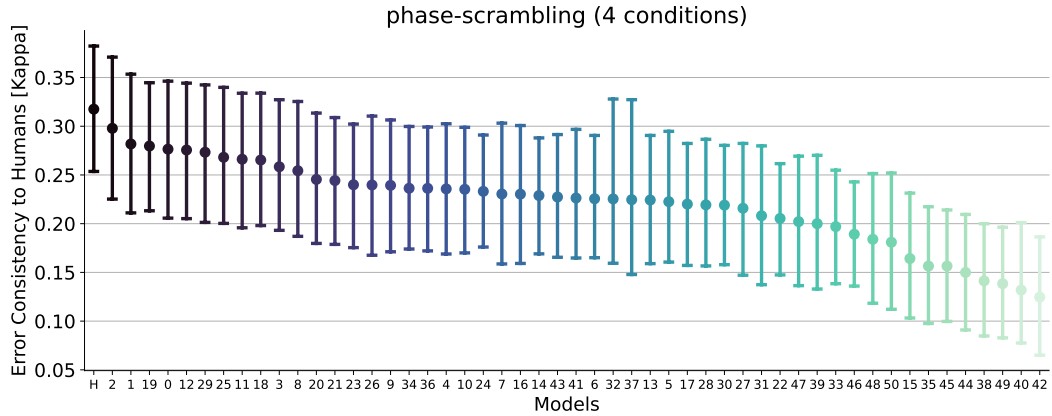

Figure 11: **ECs on phase-scrambled images.** We plot CIs for the EC to humans achieved by all models in the phase-scrambling corruption as an example, analogous to Figure 4

desired kappa (which is given by the Brain-Score data after removing the ceiling correction), the accuracy of the humans (which is given by the Model-vs-Human data) and the accuracy of the model in this condition. The latter is unknown, but bounded mathematically. We obtain the CI sizes in Figure 5 by selecting the accuracy in the middle of the bounds, which seems reasonable. Selecting the lower bounds of the accuracy leads to the most optimistic estimate of CI size, which we plot in Figure 14.

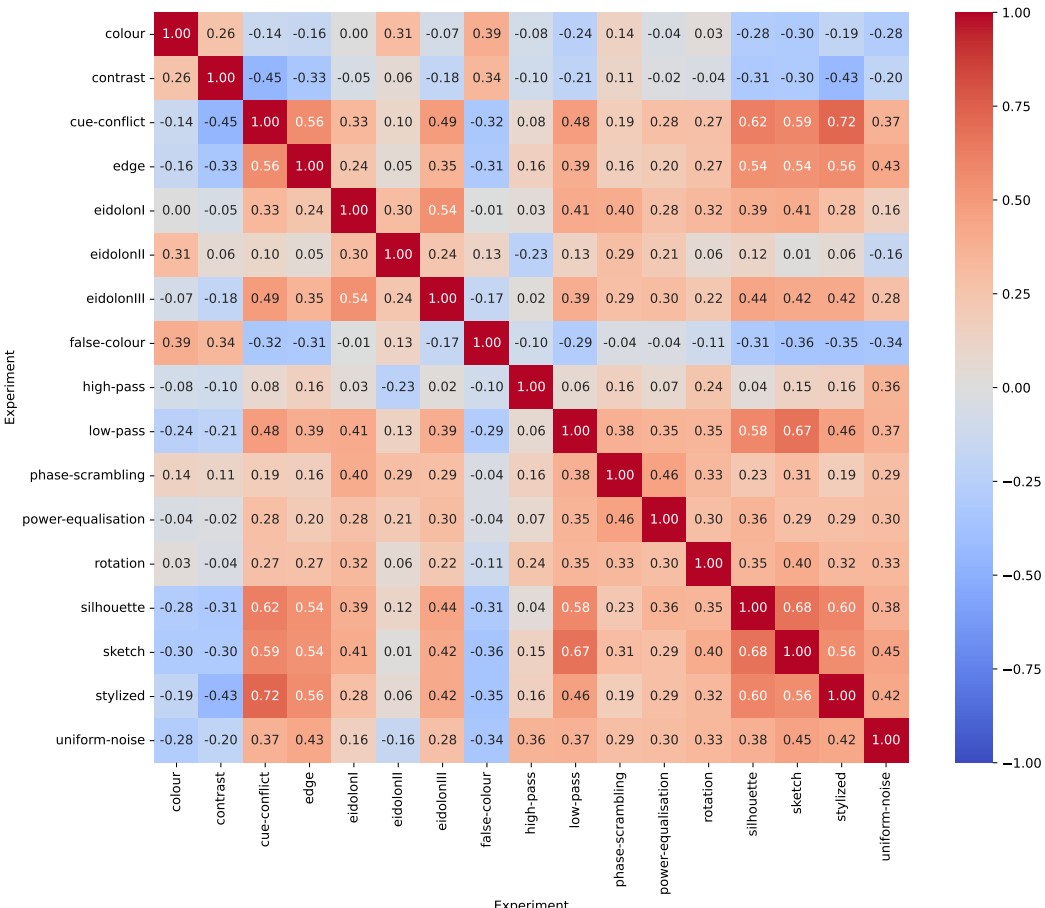

Figure 12: **Spearman Correlation between Experiments.** For all possible pairs formed by the 17 corruptions from Model-vs-Human, we compute the Spearman rank correlation between the rankings of models they imply. All rank correlations are statistically significant at $\alpha = 0.05$ (without correcting for multiple comparisons).

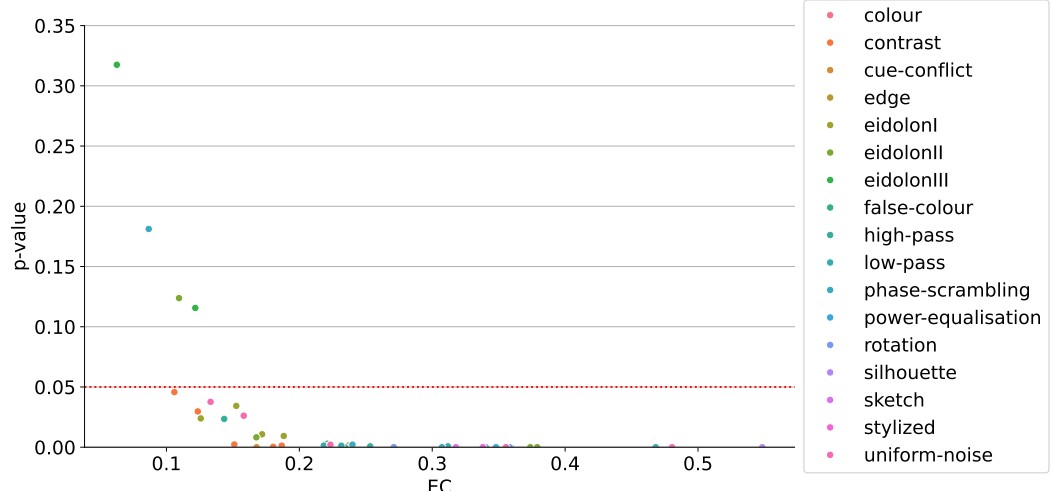

Figure 13: **p-values for the EC between CLIP and human participants.** For all experimental conditions (corruptions at a certain strength) we scatter the EC against the p-value, finding that the majority of conditions is significant, while some values close to 0 are not, which is reasonable.

| Index | Model Name | Index | Model Name |
|---|---|---|---|
| 0 | clip | 26 | vgg16_bn |
| 1 | BiTM_resnetv2_101x1 | 27 | mobilenet_v2 |
| 2 | BiTM_resnetv2_152x2 | 28 | vgg11_bn |
| 3 | BiTM_resnetv2_50x1 | 29 | resnet101 |
| 4 | resnet50_l2_eps5 | 30 | resnet152 |
| 5 | resnet50_l2_eps3 | 31 | mnasnet1_0 |
| 6 | ResNeXt101_32x16d_swsl | 32 | wide_resnet101_2 |
| 7 | BiTM_resnetv2_152x4 | 33 | resnext50_32x4d |
| 8 | BiTM_resnetv2_50x3 | 34 | resnext101_32x8d |
| 9 | resnet50_l2_eps1 | 35 | simclr_resnet50x2 |
| 10 | vit_large_patch16_224 | 36 | vgg13_bn |
| 11 | vit_base_patch16_224 | 37 | simclr_resnet50x4 |
| 12 | vit_small_patch16_224 | 38 | simclr_resnet50x1 |
| 13 | transformer_L16_IN21K | 39 | resnet50_l2_eps0 |
| 14 | densenet201 | 40 | MoCoV2 |
| 15 | resnet50_swsl | 41 | wide_resnet50_2 |
| 16 | inception_v3 | 42 | efficientnet_l2_noisy_student_475 |
| 17 | transformer_B16_IN21K | 43 | squeezenet1_1 |
| 18 | resnet50 | 44 | mnasnet0_5 |
| 19 | densenet169 | 45 | InfoMin |
| 20 | BiTM_resnetv2_101x3 | 46 | alexnet |
| 21 | resnet34 | 47 | shufflenet_v2_x0_5 |
| 22 | resnet50_l2_eps0_5 | 48 | squeezenet1_0 |
| 23 | resnet18 | 49 | MoCo |
| 24 | vgg19_bn | 50 | PIRL |
| 25 | densenet121 | 51 | InsDis |

Table 2: **Model Indices for MvH.** We map model indices in Figure 4 to their full names.

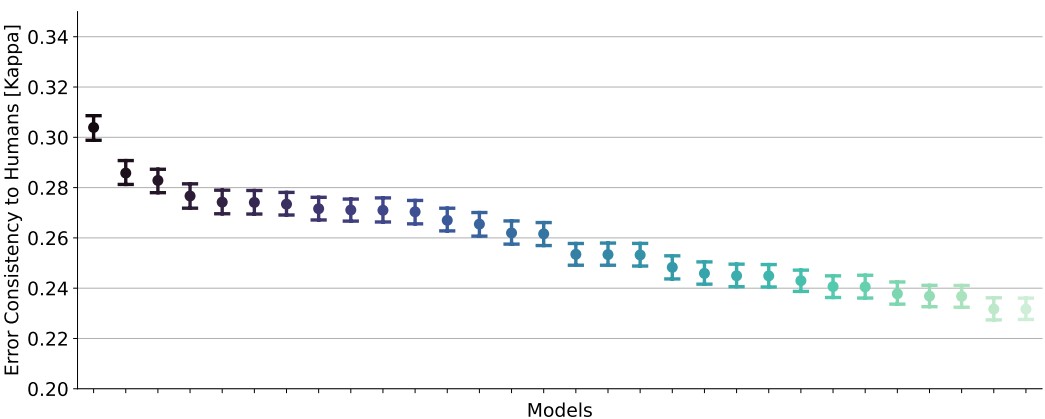

Figure 14: **Most optimistic Confidence Intervals for Brain-Score ECs.** Like in Figure 5, we plot the 95% confidence interval of the EC of the top 30 Brain-Score models, but assume the lowest mathematically possible model accuracy. The true CI size for these models probably looks more like Figure 5, but this is a lower bound on their size.

