# OpenReview forum: "Quantifying Uncertainty in Error Consistency: Towards Reliable Behavioral Comparison of Classifiers"
_NeurIPS.cc/2025/Conference — NeurIPS 2025 poster_

### Official Review · Reviewer_HmXf · 2025-07-01

**Clarity:** 3
**Significance:** 4
**Originality:** 3
**Rating:** 5
**Confidence:** 3

**Summary:**

The paper investigates error consistency (EC) and highlights several improvements in how it could be used in the Literature.
Specifically, the authors address the lack of confidence intervals commonly reported alongside the EC, propose a decomposition of the EC into interpretable components, and apply their framework to re-analyze prior studies.
Through this lens, they uncover potential statistical limitations in existing evaluations of model performance.

**Questions:**

Major points:
See Weaknesses.

Clarifications:
- Fig2.b: What would the figure look like if one included a lower boundary for -EC? Would it be the same or would the level curves be different?
- L116: On the third line of the calculations, what is $\tilde P_2$? Is it supposed to be $\tilde P^{(2)}$?
- Prop1: I believe there is a missing assumption: the independence between $\hat y^{(1)}$ and $\hat y^{(2)}$ when one is not copying the other. This is necessary to write the joint probability distribution as on L113, with a "dependent" and an "independent" parts. Also, I believe it would be _formally_ more correct to write $\tilde P^{(\cdot)}$ instead of $P^{(\cdot)}$ in the equation, as the joint is defined in terms of the underlying distributions (as the authors already do in the proof of Prop2).
- L170: What is a "condition" here? Does it correspond to the intensity of corruption?
- L187, 189, possibly elsewhere: A binomial distribution has support on $\mathbb N$, but $r$ is defined as a binary variable (L72). I think a Bernoulli distribution would fit the context better.
-  Although one can infer them, Section 3.3 does not clearly state the null and alternative hypotheses.
- L 220, 221: I do not understand this sentence and its implications. What are "prior" and "posterior" in this context? Why is the assumption of a uniform prior over the accuracies reasonable? Where does the Beta come from?

Spelling and notation:
- L70: Duplicated "and".
- I suggest replacing $P^{(\cdot)}$ with $P_\cdot$, as it would make the math more readable and less cluttered.
- I suggest replacing $y$ with $r$ in, e.g., Prop2, as it would make it clearer that, in the context of this paper, these results are applied to the "correctness" of the model, rather than the actual labels.
- L214: The symbol $\kappa$ is already used for Cohen's $\kappa$.

Open discussion points:
- Figure 3 shows how the confidence intervals are very wide even for relatively large samples. In the long run, I can imagine that, as the field advances, DNNs will become better at imitating humans, thus achieving closer EC and requiring prohibitively large samples. I would be interested in knowing the authors' take on whether EC is a good metric for this kind of experiments or whether the community should move on to something different.
- Also related to Figure 3: using the decomposition of EC in Prop2, could one "decompose" the confidence intervals into the $p_{copy}$ and $f$ CIs? That could help understand where the high-variance of EC comes from.

**Ethical Concerns:**

["NO or VERY MINOR ethics concerns only"]

**Final Justification:**

I am fully satisfied by the authors' rebuttal.

**Limitations:**

I recommend moving them from Section 3.2 to a dedicated subsection of Section 5.
.

**Paper Formatting Concerns:**

None.

**Quality:**

3

**Strengths And Weaknesses:**

## S1 - Strong motivation
The paper addresses a timely and important problem.
The motivation is clearly explained and compelling, especially thanks to  Figure 2.b.

## S2 - Novel insights
The authors' novel interpretation of EC (Prop2) helps understanding the metric, its meaning, and its limitations. I found particularly impactful the remark that a DNN with higher-than-human accuracy might be penalized in the evaluation (L139--141).

## W1 - Existing CIs in the related work
Geirhos (2020), in Section 2.2, already seems to provide a confidence interval for EC. The paper lacks a discussion of existing related work in this direction.

## W2 - Conclusions of Section 3.2
Section 3.2 raises an important point about the limited reliability of the EC when samples are relatively small, as it is common in the field (L194--198).
However, the analysis assumes a ground truth EC of 0.5, which, as shown in Appendix D, leads to the widest confidence intervals.
Thus, the authors' critique may be over-conservative, at least when applied to cases in which the EC lies further from 0.5.
I recommend (1) adding a link to Appendix D in the main text (and to other sections of the Appendix too, when relevant); (2) either moderate the strength of the conclusions drawn, or (preferably) expand Figure 3 to include a range of ground-truth EC values, indicating where existing studies and results fall along the EC spectrum.

## W3 - Informative value of the confidence intervals in Section 4.1
In Section 4.1, Figure 4 reports the average EC of individual models, averaged, among others, across humans (cf. Section 3.1).
However, given that humans already disagree among each other (EC$\approx$0.42), it is unclear to me whether the reported CIs are informative.
To clarify, I mean that the variability introduced by human disagreement may dominate the total variance and that the reported CIs may be very small compared to it, making them less useful.
This effect may become even more pronounced when other factors (such as corruption and condition) are also de-aggregated.
The paper should at least provide de-aggregated versions of Figure 4, showing error bars for the variation of EC for human and/or other factors.

---

> ### Author Rebuttal · Authors · 2025-07-31
>
> We would like to thank all reviewers for their valuable feedback. We very much appreciate their assessment of our work as containing a **“strong motivation”** (HmXf), being **“well-written and easy to follow”** (eJLm), offering **“novel insights”** (HmXf) and **“provid[ing] new tools”** (1TAb). It connects to previous work (SWie, 1TAb, eJLm) and adds **“a major theoretical result”** (1TAb) which improves our understanding of the popular Error Consistency metric. We have implemented most of the reviewers’ suggestions in the newest version of the manuscript, and hope our responses will clarify all remaining points. The reviewers’ feedback has greatly improved the submission.
> Due to the rebuttal rules this year, we cannot upload a new version of the manuscript, but can only promise to make changes in the camera-ready version, should the paper be accepted.
>
> Dear Reviewer,
>
> Thank you for reading our work with such attention to detail. We greatly appreciate the depth of your review and the specific suggestions you made, which have improved the manuscript considerably.
>
> *Q: “Geirhos (2020), in Section 2.2, already seems to provide a confidence interval for EC. The paper lacks a discussion of existing related work in this direction”*
>
> A: Thank you for this comment. You are right, Geirhos et al. indeed provided a basic variant of confidence intervals. But these confidence intervals were computed in a suboptimal way. They were based on the expected agreement $p_{exp}$ rather than the actual accuracies of the two classifiers, $p_1$ and $p_2$. This creates an ambiguity because, for example, $(p_1 = 0.5, p_2 = 0.5)$ as well as $(p_2 = 0.5, p_1 = 0.99)$ both produce the same expected agreement of $p_{exp}$ = 50% but different confidence interval widths. Moreover, these confidence intervals were necessarily centered around 0 because they were constructed only for independent observers. Our approach now enables calculating confidence intervals by making use of the observed accuracies, and for two dependent observers. We will add a section discussing this important difference in the paper. We appreciate your suggestion, which clearly improves our paper by making the difference to prior work more explicit!
>
> *Q: “The analysis assumes a ground truth EC of 0.5, which [...] leads to the widest confidence intervals. I recommend (1) adding a link to Appendix D in the main text [...]; (2) either moderate the strength of the conclusions drawn, or (preferably) expand Figure 3 to include a range of ground-truth EC values, indicating where existing studies and results fall along the EC spectrum”*
>
> A: Thank you for this actionable suggestion. We agree that this would improve the manuscript and have implemented (1) and (2). In fact, the EC values in most existing studies hover around 0.2 to 0.4, at which point the confidence intervals are very similar to the ones depicted, only the lower bound changes a bit. Unless you insist, we would stick with depicting the largest CIs, since we feel like otherwise we would be downplaying the importance of the result: Since we want to be a voice of caution, displaying the worst case scenario seems reasonable. We are more explicit about this now, though. Does this resolve your issue?
>
> *Q: “In Section 4.1, Figure 4 reports the average EC of individual models, averaged, among others, across humans”*
>
> A: You are absolutely correct, we are aware of this issue and delighted that you noticed it. In fact, an earlier draft of the manuscript included the figure you requested in the appendix, but we chose to exclude it out of fear that it would confuse less astute reviewers. Individual observers do indeed have somewhat different values, but our main message is not affected, and this is a consequence of  the specific experiment itself, not our methodology. We have now included such a figure. However, the CIs in figure 4 are informative in the sense that if one accepts the paradigm, these are conservative CIs, and they are still very large: Every other choice of aggregation would make confidence intervals even larger.
>
> We have numbered your clarifications 1 to 7:
>
> C1: Yes, the figure would indeed look different, as the upper and lower bounds are not symmetric. In the context of the experiments by Geirhos et al, the lower bound is not of much interest because all values are positive, but we now include such a figure in the appendix.
>
> C2: Yes, thank you for catching this typo.
>
> C3: Yes, we now make the independence assumption more explicit. Additionally, we changed the notation in the proof to show underlying rather than observed distributions. Technically, both variants are correct here because, in this proof, both distributions are identical. We nevertheless changed it in line with your suggestion, which we appreciate because this change improves clarity.
>
> C4: Yes, exactly. By “condition” we mean one intensity level within a corruption, so for 3 corruptions with 2 intensity levels each, you’d have 6 conditions.
>
> C5: You are absolutely correct, thanks for the suggestion, we call it a Bernoulli distribution now.
>
> C6: Good point, we explicitly state the Null hypothesis now.
>
> C7: Thank you for the question, we may have glossed over some details here and assumed an unreasonable familiarity with Bayesian statistics. The problem here is the following: Our observers had a certain accuracy, the uncertainty of which depends of course on the number of trials (i.e. if you get 6/10 trials right, I’m less certain that your true latent accuracy is 60% than when you got 600/1000 right). When we model the independent observers, we need to account for our uncertainty about their accuracy, so instead of assuming some fixed accuracy, we draw it from a distribution, which is derived like this: We assume a binomial observer, i.e. an observer that draws each response to a sequence of binary questions from a Bernoulli distribution with an (unknown, ground-truth) parameter $p$, so the whole of its responses were drawn from a binomial distribution. Initially, we know nothing about the observer, so our prior, capturing our knowledge about $p$, is uniform over {0,1} (which is commonly considered the most conservative prior, so is adequate here). After having observed $\alpha$ incorrect responses and $\beta$ correct responses, the posterior distribution $P(p | data) = \frac{P(data | p) P(p)}{P(data)}$ is given by $Beta(\alpha, \beta)$ (because the Beta distribution happens to be the conjugate prior for the binomial distribution, i.e. is the closed-form solution for the posterior, see https://en.wikipedia.org/wiki/Conjugate_prior, where this is the first example). In line 221, there is actually a typo, because we accidentally inverted $\alpha$ and $\beta$, which we have now fixed - thank you for directing our attention here!
>
> Thank you for the spelling- and notation-suggestions, we have implemented most of them. As exceptions, we still use $\kappa$ to denote EC, because it is indeed Cohen’s $\kappa$ (specifically, over the correct / incorrect responses). For this reason, we also kept responses $y$ instead of correctness $r$ in proposition 2 — where we now improve clarity by making this explicit thanks to your suggestion. Lastly, we also kept the superscript notation $P^{(1)}$ instead of $P_1$ to be consistent with the notation for responses, where we have a superscript j for the classifier and a subscript i for the trial, $y^{(j)}_i$. This is not our invention but has been done in other papers as well (e.g., in Gwet, 2016). Having said that, if you insist, we will make this change for the final manuscript version.
>
> Regarding your discussion points (which are great points, so we partially address them now in the discussion section):
> 1. Yes, to achieve a more precise resolution with EC, we would need way more trials, which will eventually be prohibitively expensive for human observers. A possible remedy would be to design experiments cleverly, by selecting stimuli that bring out the differences efficiently. Also, EC can not only be used for comparisons between humans and DNNs, but also to compare different DNNs with each other, in which case the slow convergence is less of a concern. As multimodal models become increasingly popular and we are no longer limited to pure classification outputs, more sophisticated methods of comparison will have to be developed, but for the near future, most (M)LLM evaluations employ multiple-choice paradigms, to which EC is applicable.
>
> 2. This is an excellent question and suggestion. Indeed, given that EC can be separated into two factors, copy probability and accuracy-mismatch term, the uncertainty can be similarly decomposed. We are currently working on this based on standard error propagation methods, and preliminary results support our previous recommendations (e.g., that the accuracies of the classifiers should ideally be held close to 50% to maximize the information gain about the copy probability). Although this could be an interesting addition, we fear it goes beyond the scope of the current manuscript. We would defer a thorough analysis of this uncertainty decomposition to future work.

---

### Official Review · Reviewer_eJLm · 2025-07-01

**Clarity:** 3
**Significance:** 2
**Originality:** 2
**Rating:** 3
**Confidence:** 4

**Summary:**

Error consistency quantifies whether two classifiers make consistent decisions. Geirhos et al (2020) proposed to use Cohen’s Kappa (Cohen, 1960), a popular metric in behavioral science, to evaluate the error consistency of deep networks and humans. Previous studies focused on reporting the mean of the Cohen’s Kappa. The variability and the confidence interval associated with Cohen’s Kappa have been ignored. This paper proposed a procedure to compute the confidence interval of Cohen’s Kappa using bootstrapping, and applied it to evaluate two benchmark datasets. The paper also proposed a generative model of classifier consistency, which might allow one to perform simulations to decide on the sample sizes that would be needed in the experiments for achieving certain statistical power.

**Questions:**

The paper reported results on testing whether the EC between a network and humans was substantially different from that between two independent binomial observers in Fig. 7. While this test is useful, its interpretation is also limited. Another test (potentially more useful test) would be to test if one EC is larger (or smaller) than another EC. This would allow one to make claims about which neural network exhibits behavior that is more similar to humans. Would this be possible? (Note that Figure, right panel, seems to be imply that this test is already done between the different network models, but how that was done was not described in Section 3.3.)

Line 135-144 mentioned that it would be important to ensure that human responses have uniform marginal distributions. Practically, what would be an effective procedure for achieve this?

Can the authors please explain in more detail how the bootstrapping procedure in Section 3.1 was applied in oder to take into account that each human subject likely exhibit different behaviors and thus might have Cohen’s Kappa to networks?

**Ethical Concerns:**

["NO or VERY MINOR ethics concerns only"]

**Final Justification:**

I would like to thank the authors for their detailed response during the rebuttal. The rebuttal clarified a few points. The authors acknowledged the error in the interpretation of Fig 1. It was originally claimed that because the error bar was substantial, the gap between network v.s. human and human v.s. human should shrink. Now the authors agreed that that interpretation was incorrect. While the authors considered this correction would not cause a re-interpretation of the overall contribution, I think it decreased the significance of work. That is, if the original interpretation in Fig. 1 were to be correct, that would be a substantial result. Now with the correction, we only know some of model comparison results are inconclusive due to the large error bar. While this is useful to know, it is not a substantial new insight. It remains

After reading the authors’s response on the assumption of theory, I still have concerns about it. I understand that if classifier 2’s behavior can be described as the mixture of the distribution of classifier 1 and its own marginal, the math will go through. However, the behavior of classifier 2 can be dependent on that of classifier 1 without a component that “copies” the response of classifier 1. The dependence between the two classifiers can take other form. I believe, to fully understand this question, the conditional distribution between the two classifiers need to be modeled and linked to EC. So for this point, I disagreed with what was claimed in the paper about the generality of the theoretical results.

I noticed reviewer SWie shared some similar concerns.

I do agree that the finding that the interpretation of EC depends on the accuracy of the two classifiers is a useful result, which casts doubts on the usefulness of EC in practice and motivate more carefully designed experiments. I also agreed that it’s useful to have confidence intervals. However, this contribution is rather incremental. in particular Geirhos et al 2020 also reported confidence interval. I only became aware of this from reading the review of Reviewer HmXf. The authors claimed the method in Geirhos et al 2020 is suboptimal.

Overall, I felt that, while the paper contains some useful contributions, substantial concerns remain. I increased my overal ratings from 2 to 3, yet remain overall negative about the paper.

**Limitations:**

One of the limitations of the work was acknowledged, i.e., a lack of analytical solutions.

**Paper Formatting Concerns:**

No formatting concerns.

**Quality:**

2

**Strengths And Weaknesses:**

###Strengths:

The paper is well-written and is easy to follow.

The proposed method for computing the confidence interval of Cohen’s Kappa is reasonable, and is a nice supplement to the work of Geirhos et al (2020).

Some of the limitation of Cohen’s Kappa is highlighted. For example, the upper bound of Cohen’s Kappa strongly depends on the difference in the accuracy of the two classifiers. This raised some concerns on prior results.


###Weaknesses:

The work is fairly incremental. While the ability to compute a confidence interval is useful, the proposed method based on bootstrapping is fairly standard. This by itself does not seem to be justified for a stand-alone contribution at NeurIPS.


The applications in Section 4 did not reveal substantial new insights.

The interpretation of the main results in Figure 1 is problematic. It was claimed that because the error bar was substantial, the gap between network v.s. human and human v.s. human should shrink. However, it could also be the opposite, that is, the gap was actually larger than what was implied in the difference in the mean of the Cohen’s Kappa.


The analytical results in Proposition 2 and Section 3.2 seem to be based on relatively restrictive assumptions. It is unclear whether a mixture of a copy of the response distribution of classifier 1 and another marginal distribution is sufficient to model the general response distribution of classifier 2. This needs to be clarified.

---

> ### Author Rebuttal · Authors · 2025-07-31
>
> We would like to thank all reviewers for their valuable feedback. We very much appreciate their assessment of our work as containing a **“strong motivation”** (HmXf), being **“well-written and easy to follow”** (eJLm), offering **“novel insights”** (HmXf) and **“provid[ing] new tools”** (1TAb). It connects to previous work (SWie, 1TAb, eJLm) and adds **“a major theoretical result”** (1TAb) which improves our understanding of the popular Error Consistency metric. We have implemented most of the reviewers’ suggestions in the newest version of the manuscript, and hope our responses will clarify all remaining points. The reviewers’ feedback has greatly improved the submission.
> Due to the rebuttal rules this year, we cannot upload a new version of the manuscript, but can only promise to make changes in the camera-ready version, should the paper be accepted.
>
>
> Dear Reviewer,
>
> Thank you for providing valuable feedback on our work. We were delighted to read that you found our manuscript well-written and easy to follow and agree that being able to compute confidence intervals is useful. We believe that addressing your concerns has greatly improved the manuscript, especially figure 1 (see below).
>
> *Q: “The work is fairly incremental”*
>
> A: You are right to point out that bootstrapping to generate confidence intervals would not be novel. However, our contributions go beyond that: At the moment, there is influential work that uses EC to draw widely believed conclusions about behavioral alignment between ML models and humans. But the metric itself is still poorly understood. At this point, non-incremental, “disruptive” work (e.g. proposing yet another poorly understood metric) would probably not advance the field. Instead, we believe it to be crucial to improve interpretations of the existing measure. The novel contribution we make lies in presenting a well-interpretable, generative model. This allows understanding EC better (as a scaled copy-probability) and, as a practical application, to predict the width of confidence intervals in future experiments (as agreed upon by other reviewers). This will enable future work to interpret and use the metric correctly. Without our work, there is a real risk of people overinterpreting differences in rankings made with error consistency and drawing wrong conclusions.
>
> *Q: “The interpretation of the main results in Figure 1 is problematic. It was claimed that because the error bar was substantial, the gap between network v.s. human and human v.s. human should shrink. However, it could also be the opposite, that is, the gap was actually larger than what was implied in the difference in the mean of Cohen’s Kappa.”*
>
> A: Thank you very much for pointing this out, you are absolutely correct. The phrasing in Figure 1 was indeed wrong—be assured that we never interpret our results erroneously in the text; this was a graphical error in how we represent our results in Figure 1, not an error of substance. What we meant to say was of course that the uncertainty in the measurement is revealed, so the likelihood that the gap was found by chance increases. Thus, our main message does not change. Of course, we have now fixed the figure, to say “gap size uncertain”, and apologize for the mistake. Thanks again for raising this concern!
>
> *Q: “The analytical results in Proposition 2 and Section 3.2 seem to be based on relatively restrictive assumptions. [...] This needs to be clarified.”*
>
> A: Thank you for bringing this point to our attention, we were indeed not sufficiently clear. Actually, these assumptions are not restrictive: Since we place no constraints of any kind on the marginal distribution (e.g. we do not enforce it to have a certain shape), any overall distribution attainable for observer 2 can be represented as a weighted combination of the marginal of observer 1 and this latent marginal of observer 2; its whole purpose is to “cancel out” the deviation from the empirical marginal introduced by the marginal of observer 1. The misunderstanding might arise from the fact that when we talk about the distributions, we mean the (binary) distributions over correctness here, not the distributions over responses. We have now revised this point in the manuscript . Does this clarify our reasoning?
>
> *Q: “Another test [...] would be to test if one EC is larger (or smaller) than another EC. [...] Would this be possible?”*
>
> A: Thank you for this excellent question! We agree that testing the significance of a difference in EC-values is even more relevant than just rejecting a null-hypothesis of independence. Indeed, doing this is possible with our methodology. In the simple case with full data, we have responses to N images from humans and two models. We can then bootstrap jointly from these responses, compute ECs for both human-machine pairings, and compute a difference. Doing so for each bootstrap run, we obtain a distribution over those differences, which we can then evaluate as usual. We are currently looking into this and will add an example of such a calculation to our paper. Moreover, with our generative model, we can also evaluate differences even without access to the trials themselves, by first using the model to generate sequences of trials and then running the bootstrapping. Thank you very much for raising this point; discussing and answering it has made our submission stronger.
>
> *Q: “Practically, what would be an effective procedure for achieving [uniform marginal distributions of human responses]?”*
>
> A: Thank you for this question. What we had in mind was to control the difficulty of trials for humans by standard procedures, e.g. running pilot experiments to find stimuli sets on which human participants achieve 50% accuracy, or using adaptive procedures like staircase paradigms. Alternatively, there are various standard techniques in psychophysics that are used to modulate difficulty of fixed stimuli, such as decreased presentation time or using masking patterns. In any case, controlling the marginal distribution of one of the two classifiers is sufficient. We have elaborated on this point in section 2 of the manuscript. Does this answer your question?
>
> *Q: “Can the authors please explain in more detail how the bootstrapping procedure in Section 3.1 was applied?”*
>
> A: Thank you for raising this point. We have amended the manuscript to elaborate further on the details of bootstrapping. In Section 3.1 specifically, we apply the bootstrapping approach to the data from [1], which is a bit more complex than a standard experiment: Essentially, they collected multiple trial-sequence-pairs for every pair of observers. One might be tempted to concatenate all these sequences to then calculate one final kappa for each pair of observers. However, since the sequences were obtained from experiments with differing difficulties, doing so would overestimate the EC. So instead, one has to average the ECs themselves. When we bootstrap data, we of course have to follow the same procedure to obtain correct confidence intervals. Does this explanation sufficiently clarify our bootstrapping procedure?
>
> We hope that our responses addressed your concerns satisfactorily, perhaps allowing you to reconsider the score you have given our submission. Of course, we would be very happy to engage in further discussion.
>
> [1] Geirhos, R., Narayanappa, K., Mitzkus, B., Thieringer, T., Bethge, M., Wichmann, F. A., & Brendel, W. (2021). Partial success in closing the gap between human and machine vision. Advances in Neural Information Processing Systems, 34, 23885-23899.

---

> > ### Comment · Reviewer_eJLm · 2025-08-04
> >
> > Thanks for your replies to my concerns!  The response addressed some of my concerns.
> >
> > Some follow-up questions:
> >
> > (1) what were the new insights from the application of the proposed method? This was one of the weaknesses I raised but it was not explicitly addressed in ther response.
> >
> > Related to this, I thought initially one of the main implications was that applications of this method showed a shrinkage in the gap (Fig. 1). Now given the authors agreed that Fig. 1 was a mis-interpretation/mis-representation of the results, I wonder what the new things that we learned from these empirical results.
> >
> >
> > (2) I am still uncertain about the assumptions underlying the main theoretical results in the paper. In the response, the authors claimed that the theory is general. However, according to the statement in propositions 1& 2, the distribution of classifier 2 has to be a mixture of the distribution of classifer 1 and its own marginal. The assumptions simplify the math, but this formulation doesn’t seem to be able to capture a general joint distribution of the two classifiers. Can you please clarify this issue further?

---

### Official Review · Reviewer_1TAb · 2025-07-02

**Clarity:** 4
**Significance:** 3
**Originality:** 4
**Rating:** 5
**Confidence:** 4

**Summary:**

This paper presents a study of error consistency (EC), a standard metric to measure the behavior alignment between ML models and human observers.

A major theoretical result is a new computational model, which frames EC as the probability that one classifier copies the other, multiplied by a factor originating from the accuracy mismatch. This result serves two purposes.

First, it provides an alternative interpretation of EC, offering a clearer picture that it can be decomposed into the two factors of the copying probability and mismatching accuracy.

Second and perhaps more importantly, it offers a means of computing confidence interval for EC, a missing component in previous studies that only provides a point estimation and hence may not be reliable. More importantly, while bootstrapping can be trivially applied for estimating confidence intervals, the technique from their Generative computational model can be useful for planning future experiments, as it helps to determine the number of trials needed to obtain statistically reliable results.

The authors applied their new methodology to re-analyze the results from major benchmarks, including Model-VS-Human and Brain-Score.

**Questions:**

I don't have any questions.

**Ethical Concerns:**

["NO or VERY MINOR ethics concerns only"]

**Final Justification:**

By reading the other reviewers' comments, there are concerns about the significance of the contribution and generality of the theoretical result. I am unfortunately not knowledgeable enough in this area to evaluate those aspects.

I rate the paper as acceptance mainly because the paper is well-written and I learned some new interesting insights, and that I believe that the general (non-expert) audience of the conference can benefit the same from this paper.

**Limitations:**

Yes

**Quality:**

4

**Strengths And Weaknesses:**

Strengths:

- The paper provides new insights into the EC metric with the decomposition into the copy probability and the accuracy mismatch. This can be important for understanding what is truly being measured by this metric.
- The paper provides new tools for computing confidence intervals for EC, a missing piece in previous works.
- The paper revisited previous experiments using their algorithm, and revealed new insights (e.g., a larger number of trials is essential to obtain statistically significant results).

Weaknesses:

I don't have any particular concerns. One suggestion is to put the algorithms in Sec 3.1 and Sec 3.2 in an algorithm box to improve on clarity.

Overall, the paper has a nice flow that started with an in-depth analysis of the metric itself, revealing insights into what it is truly measuring; then went into the key objective of  calculating confidence intervals, which makes use of the aforementioned analysis. While I don't have experience in this area (hence cannot comment on related work), the paper does appear to offer a quite solid study of EC, that can benefit future studies in using it as a tool for measuring machine vs human consistency and beyond.

---

> ### Author Rebuttal · Authors · 2025-07-31
>
> We would like to thank all reviewers for their valuable feedback. We very much appreciate their assessment of our work as containing a **“strong motivation”** (HmXf), being **“well-written and easy to follow”** (eJLm), offering **“novel insights”** (HmXf) and **“provid[ing] new tools”** (1TAb). It connects to previous work (SWie, 1TAb, eJLm) and adds **“a major theoretical result”** (1TAb) which improves our understanding of the popular Error Consistency metric. We have implemented most of the reviewers’ suggestions in the newest version of the manuscript, and hope our responses will clarify all remaining points. The reviewers’ feedback has greatly improved the submission.
> Due to the rebuttal rules this year, we cannot upload a new version of the manuscript, but can only promise to make changes in the camera-ready version, should the paper be accepted.
>
>
> Dear Reviewer,
>
> Thank you for your review. We were delighted to read that you believe our work **"provides new insights"** and that our new model is a **"major theoretical result"**. We welcome your suggestions to provide supporting explanations. For that, we have written explicit algorithms for the procedures we describe in sections 3.1 and 3.2, which we would include in the camera-ready version upon acceptance. This does indeed facilitate the understanding of the bootstrapping procedure, which other reviewers also asked about, so thank you for the valuable suggestion.

---

### Official Review · Reviewer_SWie · 2025-07-11

**Clarity:** 2
**Significance:** 3
**Originality:** 1
**Rating:** 4
**Confidence:** 3

**Summary:**

This paper addresses the issue of quantifying uncertainty in Error Consistency (EC), which is a standard metric used to measure the behavioral alignment between ML models and human behaviors. The authors improve the standard EC in two ways:
- They demonstrate how to obtain confidence intervals for EC using a bootstrapping technique, and derive significance tests for EC.
- They propose a new generative model that simulates the experiment and estimates the resulting confidence intervals. This model can be used to provide practical guidance on the number of trials required for experiments.

The paper applies this methodology to existing experimental results. The authors emphasize that their method enables researchers to design experiments that can reliably detect behavioral differences and fosters more rigorous benchmarking of behavioral alignment.

**Questions:**

- What are the takeaways we can get from analyzing with the confidence intervals but cannot with point estimates?
- It would be better if the authors can offer more discussion over the definition of EC and expand the results to other metrics besides Cohen's $\kappa$.

**Ethical Concerns:**

["NO or VERY MINOR ethics concerns only"]

**Final Justification:**

Thank the authors for their responses during the rebuttal.

I found that most of my concerns are the same with reviewer eJLm's. The most important contribution of this paper is the uncertainty quantification of EC, but I am still concerned that the method used by the authors is not enough. When you have access to the trail-level data, it is natural to quantify the uncertainty using bootstrap. The new method proposed by the authors applies to the case before the actual data is collected. However, it largely depends on the interpretation of EC and the current manuscript lacks validation on this.

The authors mentioned that they will add a result showing the match between simulated results and empirical results. This analysis can be important for validating the model. Based on the commitment to include this validation, I have raised my score from 3 to 4.

**Limitations:**

yes

**Paper Formatting Concerns:**

No major formatting issues are found.

**Quality:**

2

**Strengths And Weaknesses:**

Strengths:
- The paper is clearly motivated: the measure of EC is inherently noisy, so quantifying uncertainty is necessary when analyzing the results.
- The discussion with theoretical issues with EC offer a solid theoretical ground for the paper.
- The solution (bootstrapping) offered by the paper is concrete and pragmatic for calculating confidence intervals for empirical EC values. The proposed generative model can be useful for practitioners to determine the sample size of experiment.
- The re-analysis of existing experiments (Model-VS-Human and Brain-Score) provides empirical evidence for the issues of uncertainty and demonstrates the utility of their methodology.

Weaknesses:
- The assumptions of bootstrapping are not discussed, such as the sample need to be representative, which can be questionable for some scenarios of error consistency between human and ML classifiers. For example, when the sample of human is too small and cannot represent the population of interest.
- The paper simply applies bootstrapping to EC, leaving the choice of parameters in bootstrapping that could impact the results dramatically.
- The motivation for the definition of EC is not enough. Why is EC modeled with $p_{copy}$? Is it the only choice?

---

> ### Author Rebuttal · Authors · 2025-07-31
>
> We would like to thank all reviewers for their valuable feedback. We very much appreciate their assessment of our work as containing a **“strong motivation”** (HmXf), being **“well-written and easy to follow”** (eJLm), offering **“novel insights”** (HmXf) and **“provid[ing] new tools”** (1TAb). It connects to previous work (SWie, 1TAb, eJLm) and adds **“a major theoretical result”** (1TAb) which improves our understanding of the popular Error Consistency metric. We have implemented most of the reviewers’ suggestions in the newest version of the manuscript, and hope our responses will clarify all remaining points. The reviewers’ feedback has greatly improved the submission.
> Due to the rebuttal rules this year, we cannot upload a new version of the manuscript, but can only promise to make changes in the camera-ready version, should the paper be accepted.
>
> Dear Reviewer,
>
> Thank you for taking the time to review our paper. We appreciate that you found our paper to be **clearly motivated**, and believe that our **generative model can be useful**. Based on your feedback, we have amended the manuscript to elaborate further on the details of bootstrapping. In summary, we follow standard procedures as suggested by e.g. [1], which are very robust to parameter choices, see below for a more detailed explanation.
>
> *Q: “The assumptions of bootstrapping are not discussed. [...] For example, when the sample of humans is too small and cannot represent the population of interest”*
>
> A: While you are right that a fundamental requirement for our approach to work is a well-conducted study, like the one by Geirhos et al. (2021), our method is concerned with something else: Once we have an empirical measurement for one pair of observers, how can we estimate the uncertainty in that measurement? The point of bootstrapping is orthogonal to the question whether the sample of human observers is representative, which is, of course, also an interesting question, but not one we address.
>
> *Q: “Why is EC modelled with $p_{copy}$? Is it the only choice?”*
>
> A: Thank you for this question. In principle, it is possible that other generative models of EC exist, but our model based on $p_{copy}$ has the significant advantage of being very interpretable, providing us with an intuition for what Error Consistency really means. This is itself a contribution of our work: Previously, the metric was not well understood, so there was a risk of people using it incorrectly. Our model also allows us to simulate trial data, which is very useful for planning experiments and for the analyses we conducted here (e.g. we can estimate confidence intervals for Brain-Score models without access to most of their data). Does this answer sufficiently address your concerns? The definition of EC, which you also asked about, is outside of our control: It is, as you write, an established metric already, which measures a quantity of interest that is not captured by the percentage of correct answers given by two classifiers.
>
> *Q: “What are the takeaways we can get from analyzing with the confidence intervals but cannot with point estimates?”*
>
> A: When ML models are evaluated regarding their behavioral alignment with humans, confidence intervals are crucial for proper evaluation of the measure. For example, say we have two models, each with EC = 0.5 (close to the alignment between humans). Critically, if the confidence interval of one was very wide, say [0.3, 0.7], and that of the other was very narrow, say [0.49, 0.51]. Our interpretations are substantially different in the two cases: For the first, we could assume that the high EC value came from random fluctuations (especially in our context, where multiple classifiers are evaluated and the best one is considered), and it is not ruled out that the ground truth is that this first model has a relatively low alignment. On the other hand, we could be reasonably certain that the second model is well aligned with human behavior; the results are certainly not due to measurement noise. Thus, analyzing EC with confidence intervals substantially changes interpretations of point estimates here: They tell us whether observed differences in the point estimates are meaningful or not.
>
>
> **Details of Bootstrapping**
>
> Thank you for pointing out that our explanation of bootstrapping may have been too short. We have elaborated more on this point now, in summary: Given two sequences of length N from two observers with indices ${0,...,N-1}$, we first draw N indices randomly, with uniform probability and replacement. We then obtain bootstrapped trials by selecting the observers’ trials at the chosen index positions, yielding two new sequences of trials. From these new sequences, we can calculate new estimates of EC. Repeating this procedure multiple times yields distributions of EC values, thereby quantifying the uncertainty in the measurement.
> To address your concern about parameters (which bootstrapping requires very few of), we have also added a paragraph justifying our parameter choices. Basically, there is only one variable parameter, which we typically denote as M, the number of bootstraps. While higher M is always better, we find that a value of 10,000 is sufficient, and have added a figure to the appendix which illustrates how bootstrapped EC values converge as a function of M. The other choices, such as sampling N items with replacement for a sequence of length N and drawing uniformly is how a non-parametric bootstrap is performed, see [1].
>
> [1] Efron, B., & Tibshirani, R. J. (1994). An introduction to the bootstrap. Chapman and Hall/CRC.

---

> > ### Comment · Reviewer_SWie · 2025-08-05
> >
> > Thanks for the rebuttal. It addresses most of my concerns.
> >
> > > What are the takeaways we can get from analyzing with the confidence intervals but cannot with point estimates?
> >
> > I'm still concerned with the empirical results observed in the demonstration. This variation of confidence intervals doesn't appear in the figure 4 and 5. So what are the new takeaways from the analysis with confidence interval? I agree with your claim with confidence interval but I'm just concerned whether it's a valid point with EC.
> >
> > > Why is EC modeled with $p_{copy}$? Is it the only choice?
> >
> > Thanks for clarifying. I have some follow-up questions about this. Is this model necessary for using bootstrapping on EC? How largely does this choice of theoretical model impact the confidence intervals?
> >
> > Thanks,

---

> > > ### Author Response · Authors · 2025-08-06
> > >
> > > Dear Reviewer SWie,
> > >
> > > thank you for the follow-up questions, we greatly appreciate the discussion.
> > >
> > > *“This variation of confidence intervals doesn’t appear in figure 4 and 5”*
> > >
> > > Thank you for the question! Maybe we should have plotted this differently: The variation does appear in these figures. Note how the confidence intervals of different models all overlap. For example, in Figure 4, model 24 has overlapping CIs with all models from 8 to 45, more than 70% of all models! It is conceivable that if these experiments were done again, this model might jump up to rank 8, or down to rank 45. It just so happens that the very best and very worst models broaden the range we have to plot, making this a bit less obvious than in Figure 1. We will strengthen this point in our manuscript. Does this clarify why obtaining confidence intervals is crucial?
> > >
> > > *“Is this model necessary for using bootstrapping on EC?”*
> > >
> > > Thank you for the question. The answer depends on which data is available. If one has access to trial-level data, our model is not necessary for bootstrapping. However, it is important to note that our model is still very helpful in this case because it offers an interpretation; a notion of what an EC of e.g. 0.5 even means, which was unclear before (See section 2).
> > >
> > > However, without access to trial-level data (for example, benchmarks that only show the results but do not provide access to the underlying trial-by-trial data), a model like ours becomes a necessary requirement to obtain sensible confidence intervals (apart from some special cases like independent observers with an error consistency of zero). Thus, we would argue that our model is very helpful in the first case (for interpreting the results), and even necessary in the second case. Thank you for raising this issue; we will address it in the final manuscript.
> > >
> > > *“How largely does this choice of theoretical model impact the confidence intervals?”*
> > >
> > > Thank you for this great question! As we pointed out above, the size of bootstrapped CIs is independent of any theoretical model, but any theoretical model has to match these “ground-truth” CIs. We have indeed validated our model against these CIs and found that the match is near-perfect. We will add a figure to show this match between empirical data and model predictions in the appendix of the final manuscript.

---

### Note · Authors · 2025-08-14

Dear reviewers,

We would like to sincerely thank all of you for your detailed and insightful feedback. We highly appreciate your effort in assessing our work. Such detailed and helpful feedback is unfortunately not a given at NeurIPS nowadays. Your constructive and thorough comments have clearly improved the quality and clarity of our paper. We very much appreciate the time you invested in our submission and hope that, with our revisions, we were able to address your suggestions to your satisfaction.

Sincerely yours,

Anonymous Authors

---

### Decision · Program_Chairs · 2025-09-17

**Decision:**

Accept (poster)

**Comment:**

This paper analyzes the behavior of error consistency (EC), a widely used metric for model–human behavioral alignment, and proposes methods to quantify its uncertainty for use in designing more reliable experimental comparisons. The contributions include: (i) bootstrapped confidence intervals and significance tests for EC, (ii) a generative model interpreting EC as a copy probability scaled by an accuracy-mismatch factor that enables experiment planning, and (iii) a python package realizing these methods. The authors re-analyze major benchmarks and show that many model-to-model comparisons are not statistically reliable despite differences being claimed in prior work.

Reviewers appreciated the strong motivation, clear writing, and the interpretability provided by the decomposition of EC, which stands to impact the use of this common metric. There were some concerns raised about novelty (bootstrapping being a standard technique), about assumptions that are made (e.g., about the two classifiers relation), and about a mistake in one figure. The rebuttal did a satisfactory job of clarifying and making the case for the contribution. The authors should follow through on all revisions they promised in the rebuttal, including validation of the model against bootstrap CIs, corrected misinterpretations, and promised expanded analyses and clearer discussion of assumptions.

Overall, this is a technically solid and practically minded critical methods paper addressing a timely gap in evaluation practice. All but one reviewer recommends acceptance. For the camera-ready, the authors should include the promised validation of their theoretical model against empirical results, clarify assumptions and relation to CIs in prior work (e.g., Geirhos et al. 2020), and add the expanded analyses (pairwise tests, broader EC ranges, disaggregated human variability) to strengthen clarity and impact.